# DELTA: Dense Efficient Long-range 3D Tracking for any video

**Tuan Duc Ngo**[*]
UMass Amherst

**Peiye Zhuang**
Snap Inc.

**Chuang Gan**
UMass Amherst

**Evangelos Kalogerakis**
UMass Amherst & TU Crete

**Sergey Tulyakov**
Snap Inc.

**Hsin-Ying Lee**
Snap Inc.

**Chaoyang Wang**
Snap Inc.

https://snap-research.github.io/DELTA/

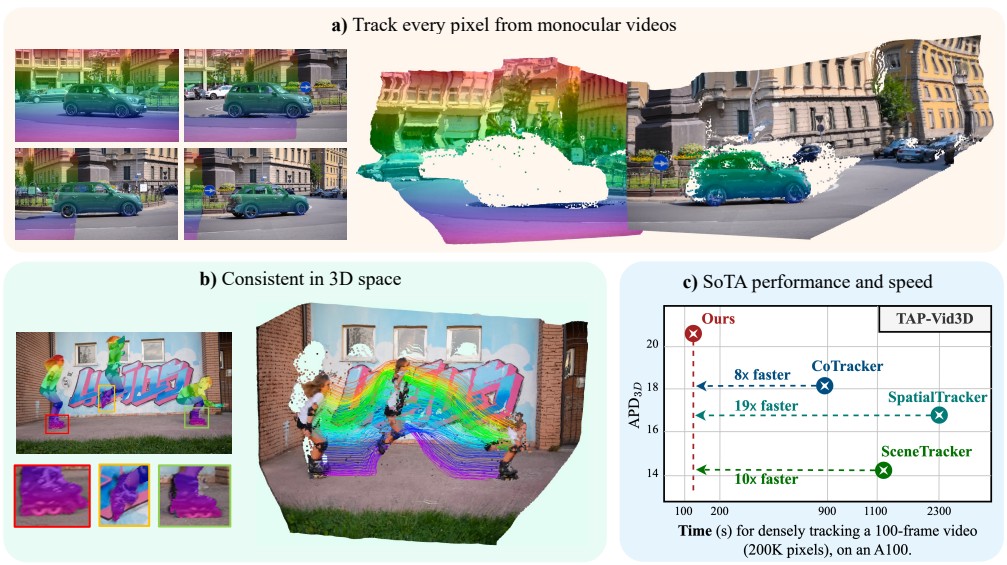

Figure 1: **DELTA** is a dense 3D tracking approach that (a) tracks *every* pixel from a monocular video, (b) provides consistent trajectories in 3D space, and (c) achieves state-of-the-art accuracy on 3D tracking benchmarks while being significantly faster than previous methods in the dense setting.

## Abstract

Tracking dense 3D motion from monocular videos remains challenging, particularly when aiming for pixel-level precision over long sequences. We introduce DELTA, a novel method that efficiently tracks every pixel in 3D space, enabling accurate motion estimation across entire videos. Our approach leverages a joint global-local attention mechanism for reduced-resolution tracking, followed by a transformer-based upsampler to achieve high-resolution predictions. Unlike existing methods, which are limited by computational inefficiency or sparse tracking, DELTA delivers dense 3D tracking at scale, running over 8x faster than previous methods while achieving state-of-the-art accuracy. Furthermore, we explore the impact of depth representation on tracking performance and identify log-depth as the optimal choice. Extensive experiments demonstrate the superiority of DELTA on multiple benchmarks, achieving new state-of-the-art results in both 2D and 3D dense tracking tasks. Our method provides a robust solution for applications requiring fine-grained, long-term motion tracking in 3D space.

## 1 Introduction

Accurately estimating motion and determining point correspondences in dynamic 3D environments is a longstanding challenge in computer vision. In this work, we aim to achieve dense 3D tracking

---

[*]Work done during internship at Snap Inc.

| Method | Dense | 3D | Long-term | Feed-forward |
|---|:---:|:---:|:---:|:---:|
| RAFT (Teed & Deng, 2020) | ✓ | | | ✓ |
| TAPIR (Doersch et al., 2023) | △ | | ✓ | ✓ |
| CoTracker (Karaev et al., 2023) | △ | | ✓ | ✓ |
| SpatialTracker (Xiao et al., 2024) | △ | ✓ | ✓ | ✓ |
| SceneTracker (Wang et al., 2024a) | △ | ✓ | ✓ | ✓ |
| DOT (Le Moing et al., 2024) | ✓ | | ✓ | ✓ |
| OmniMotion (Wang et al., 2023a) | △ | | ✓ | |
| DELTA (Ours) | ✓ | ✓ | ✓ | ✓ |

Table 1: Comparison of different types of motion estimation methods. △ denotes that the method is technically applicable to dense tracking but will be extremely time-consuming.

by establishing correspondences for **every pixel** from a given monocular video. 3D tracking provides richer insights into object trajectories, depth, and scene interactions than 2D tracking, while dense tracking captures subtle, fine-grained motions often missed by sparse methods. The task is particularly challenging due to the need to simultaneously address ill-posed 3D-to-2D projections, occlusions, camera motion, and dynamic scene changes.

The ultimate goal of tracking is to ensure both *dense* coverage and *long-term* consistency. Early efforts focused on predicting dense motion for adjacent frames or short-term sequences using optical flow (Ilg et al., 2017; Sun et al., 2018; Teed & Deng, 2020; Xu et al., 2022; Dong et al., 2023; Huang et al., 2022) and scene flow (Vogel et al., 2015; Liu et al., 2019; Yang & Ramanan, 2020), but these approaches usually struggle to capture long-term motion. In contrast, point-tracking methods (Doersch et al., 2022; Harley et al., 2022; Doersch et al., 2023; Li et al., 2024b) built correspondences over hundreds of frames but are limited to sparse pixels. Recently, hybrid approaches have emerged that attempt to combine both paradigms, yet they either rely on per-frame optical flow predictions and lack strong temporal correlation (Le Moing et al., 2024), or adopt suboptimal attention designs and cannot perform dense tracking efficiently (Karaev et al., 2023). More, advancements in depth estimation (Bhat et al., 2023; Piccinelli et al., 2024) have allowed for lifting 2D tracking to 3D (Wang et al., 2024a; Xiao et al., 2024), but these pipelines remain computationally prohibitive for dense tracking due to cross-track attention. We summarize the characteristic of these methods in Table 1.

In this paper, we introduce DELTA, **D**ense **E**fficient **L**ong-range 3D **T**racking for **A**ny video. To our knowledge, the first method capable of **efficiently** tracking **every pixel** in 3D space over hundreds of frames. We achieve efficient dense tracking using a coarse-to-fine strategy, starting with coarse tracking via a spatio-temporal attention mechanism at reduced resolution, followed by an attention-based upsampler for high-resolution predictions. Our key design choices include:

- An efficient spatial attention architecture that captures both global and local spatial structures of the dense tracks, with low computational complexity, enabling end-to-end learning for dense tracking.
- An attention-based upsampler, carefully designed to provide high-resolution, accurate tracking with sharp motion boundaries.
- A comprehensive empirical analysis of various depth representations, showing that the log-depth representation yields the best 3D tracking performance

These designs enable DELTA to capture hundreds of thousands of 3D trajectories in long video sequences within a single forward pass, completing the process in under two minutes for 100 frames—over 8x faster than the fastest existing methods, as shown in figure 1. DELTA is extensively evaluated on both 2D and 3D dense tracking tasks, achieving state-of-the-art results on the CVO (Wu et al., 2023; Le Moing et al., 2024) and Kubric3D (Greff et al., 2022) datasets both with more than **10%** improvement in AJ and $APD_{3D}$. Additionally, it performs competitively on conventional 3D point tracking benchmarks, including TAP-Vid3D (Koppula et al., 2024) and LSFOdyssey (Wang et al., 2024a; Zheng et al., 2023).

## 2 RELATED WORK

**Optical Flow** estimates motion by providing dense pixel-wise correspondences between consecutive frames. Early variational approaches (Mémin & Pérez, 1998; Horn & Schunck, 1980; Brox

et al., 2004) struggled with robustness in complex scenes with rapid motion, occlusions, and large displacements. The introduction of CNN-based methods (Ilg et al., 2017; Ranjan & Black, 2017; Xu et al., 2017; Sun et al., 2018) improved motion estimation between adjacent frames. RAFT (Teed & Deng, 2020) marked a breakthrough by leveraging 4D correlation volumes for all pairs of pixels. Follow-up works advanced this by incorporating transformers for tokenizing 4D correlation volumes (Huang et al., 2022), adopting global motion feature aggregation to improve prediction in occluded regions (Jiang et al., 2021), and framing optical flow as a matching problem with correlation softmax operations (Xu et al., 2022). While some efforts propose to apply optical flow to long-term sequences with multi-frame optical flow (Teed & Deng, 2020; Godet et al., 2021; Shi et al., 2023) or integration of point-tracking techniques (Le Moing et al., 2024; Cho et al., 2024a), they often suffer from drifting and occlusion challenges, limiting their reliability for long-term tracking.

**Scene Flow** generalizes optical flow into 3D, estimating dense 3D motion. One line of work uses RGB-D data (Hadfield & Bowden, 2011; Hornacek et al., 2014; Quiroga et al., 2014; Teed & Deng, 2021b), while others estimates 3D motion from point clouds (Liu et al., 2019; Wang et al., 2020; Gu et al., 2019; Niemeyer et al., 2019). Recent methods have improved robustness by using rigid motion priors, either explicitly (Teed & Deng, 2021b) or implicitly (Yang & Ramanan, 2021). Nonetheless, integrating scene flow methods for long sequences is under-explored.

**Point Tracking** estimates global motion trajectories in videos. Particle Video (Sand & Teller, 2008) introduced particle trajectories for long-range video motion. TAP-Vid (Doersch et al., 2022) provided a comprehensive benchmark to evaluate point tracking and TAPNet, a baseline that predicts tracking locations using correlation features. PIPs (Harley et al., 2022) revisited the concept of particle video and proposed a feedforward network that updates motion iteratively over fixed temporal windows, but ignored spatial context with independent point tracking and struggle with occlusion. Subsequent efforts addressed these limitations by relaxing the fixed-length window to variable lengths (Doersch et al., 2023) and jointly tracking multiple points and strengthening correlations between tracking points with temporal attention for temporal smoothness and spatial attention (Karaev et al., 2023). Recent approaches like SceneTracker (Wang et al., 2024a) and SpatialTracker (Xiao et al., 2024) extend point tracking to 3D by incorporating depth information, but remain inefficient for dense tracking due to computationally expensive cross-track attention. Our model builds on the strengths of these methods, but scales to full-resolution tracking.

**Tracking by Reconstructing** estimates long-range motion by reconstructing a deformation field. OmniMotion (Wang et al., 2023b) optimizes a NeRF (Mildenhall et al., 2020) representation with a bijective deformation field (Dinh et al., 2016), then extracts 2D trajectories using this bijective mapping, but suffers from instability and requires hours to optimize. Recent work with DINOv2 (Oquab et al., 2023) uses its superior semantic features to establish long-range correspondences, either with an improved invertible deformation field (Song et al., 2024) or in a self-supervised manner (Tumanyan et al., 2024). While these approaches can produce dense motion trajectories, they require per-video optimization, which is computationally expensive, and their performance on tracking benchmarks lags behind data-driven tracking methods.

We are the first feed-forward approach that performs dense 3D tracking efficiently from a long-term video. Table 1 provides a brief comparison of existing methods alongside our approach.

## 3 METHOD

**Problem setup.** We propose a method to track every pixel of a video in 3D space. Specifically, our method takes an RGB-D video as input, where the RGB frames are denoted as $\mathcal{V} \in \mathbb{R}^{T \times H \times W \times 3}$, with $T$, $H$, and $W$ representing the temporal and spatial resolution of the video, and the depth maps $\mathcal{D} \in \mathbb{R}^{T \times H \times W}$ are obtained from an off-the-shelf monocular depth estimation method. Our method then estimates dense, occlusion-aware 3D trajectories $\mathcal{P} \in \mathbb{R}^{T \times H \times W \times 4}$, where each 4D slice, $\boldsymbol{p}_{t,u,v} = (u_t, v_t, d_t, o_t)$, represents the tracking result for a pixel located at $(u, v)$ in the first frame as it moves to its corresponding 3D position in the $t$-th frame. Specifically, $(u_t, v_t)$ are the pixel coordinates in the $t$-th frame, $d_t$ is the depth estimate, and $o_t \in \{0, 1\}$ is the visibility prediction.

### 3.1 PRELIMINARY: POINT TRACKING

Our method is inspired by recent advances in 2D point tracking, most notably CoTracker (Karaev et al., 2023), which uses a transformer architecture that takes as input a set of trajectories within a

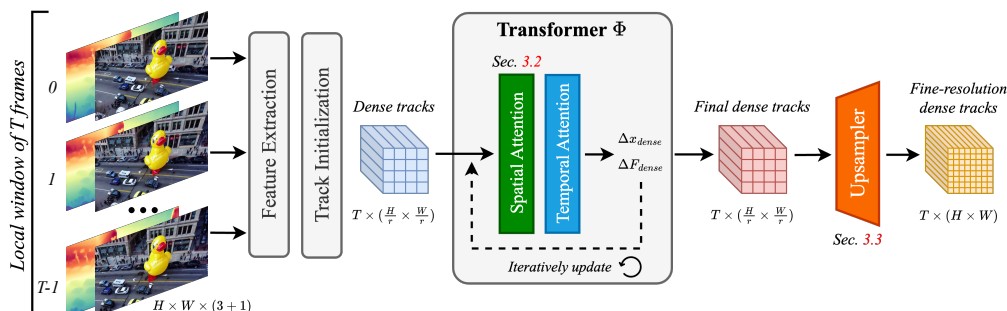

Figure 2: **Overview of DELTA.** DELTA takes RGB-D videos as input and achieves efficient dense 3D tracking using a coarse-to-fine strategy, beginning with coarse tracking through a spatio-temporal attention mechanism at reduced resolution (Sec. 3.1, 3.2), followed by an attention-based upsampler for high-resolution predictions (Sec. 3.3).

fixed temporal window and iteratively predicts the position offsets and visibilities of points based on features extracted around their current locations. It is extended by SceneTracker (Wang et al., 2024a) and SpatialTracker (Xiao et al., 2024) to include 3D-aware features for 3D point tracking.

Specifically, the transformer processes a set of initial trajectories, denoted as $\{P_i\}$, where $i$ is the trajectory index, and $P_i = [\boldsymbol{p}_1^i, \boldsymbol{p}_2^i, \cdots, \boldsymbol{p}_T^i]$, with $\boldsymbol{p}_t^i = (u_t^i, v_t^i, d_t^i, o_t^i)$ being the 3D location and visibility of the point associated with the $i$-th trajectory at the $t$-th frame. The initial values $(u_t^i, v_t^i, d_t^i, o_t^i)$ are typically initialized as $(u_1^i, v_1^i, d_1^i, 1)$, assuming that each point starts from the same location in the first frame and is visible at the beginning. We iteratively repeat the following:

**Extract token features.** Each input trajectory is represented by a list of tokens $G^i = [G_1^i, G_2^i, \cdots, G_T^i]$, each token $G_t^i$ encodes position, visibility, appearance and correlation of the trajectory at $t$-th frame:

$$G_t^i = [F_t^i, C_t^i, D_t^i, o_t^i, \gamma(\boldsymbol{x}_t^i - \boldsymbol{x}_1^i)] + \gamma_{pos}(\boldsymbol{x}_t^i) + \gamma_{time}(t), \tag{1}$$

where each term represents:

• *Track features* $F_t^i$ represents the appearance of the point to be tracked. It is initialized by sampling from the feature map at the starting location of the trajectory in the first frame, and will be updated by the transformer network.

• *Correlation features* $C_t^i$ are computed by comparing track features to image features around the current estimated track location, similar to previous optical flow and point tracking methods (Teed & Deng, 2020; Harley et al., 2022; Karaev et al., 2023). Additionally, we follow LocoTrack (Cho et al., 2024b) by including local 4D correlation, which utilizes all-pair correspondences to establish more precise and bidirectional correspondences, enhancing robustness against ambiguities.

• *Depth correlation* $D_t^m$ is calculated as the difference between the current estimated depth and the depth queried from the depth map around the estimated track location.

• *Spacetime positions* $\gamma_{pos}$ and $\gamma_{time}$ are the positional embedding of the input position $\boldsymbol{x}_t^i = (u_t^i, v_t^i, d_t^i)$ and time $t$, respectively.

• *Relative displacement.* It is also beneficial to separately encode the relative displacement of the points by computing $\gamma(\boldsymbol{x}_t^i - \boldsymbol{x}_1^i)$ where $\gamma$ represents positional embedding.

**Iteratively apply transformer.** The trajectory tokens will then be iteratively updated by applying a transformer $\Phi$. Each iteration computes updates for point positions and track features, *i.e.*

$$\{\Delta \boldsymbol{x}_t^i\}, \{\Delta F_t^i\} = \Phi(\{G^i\}). \tag{2}$$

Visibility $o_t^i$ is predicted only in the last iterative step when the accurate location has been estimated.

**Spatial-temporal transformer architecture.** The architecture consists of *temporal* attention (self-attending within the same track) and *spatial* attention (cross-track within the same frame).

**Limitations in dense tracking settings.** By default, CoTracker is trained and tested in sparse tracking settings, where the number of tracks $N$ is kept low ($< 10^3$). In dense settings, where the total number of tracks is $N = H \times W$ (the order of $10^5$-$10^6$), the spatial attention becomes a bottleneck due to limitations w.r.t. its computational cost and spatial granularity explained below.

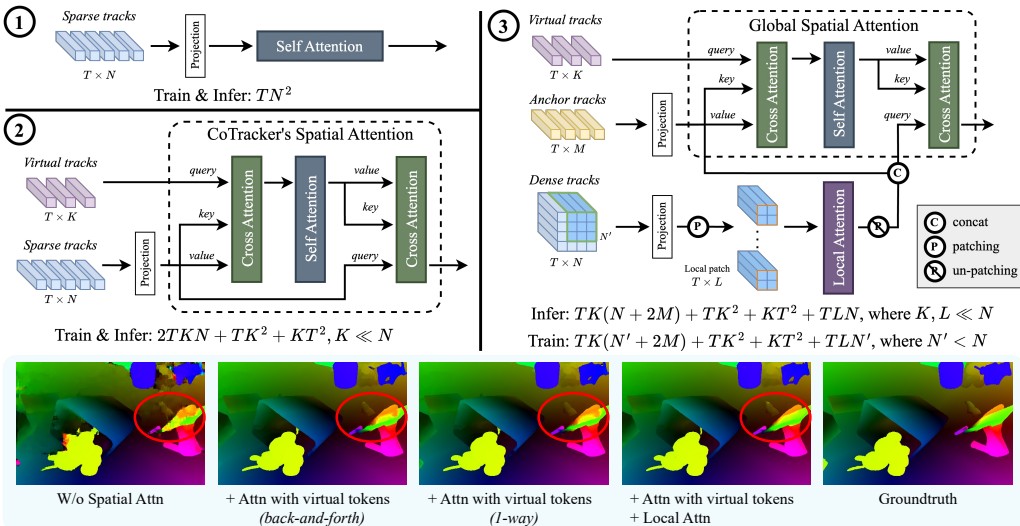

Figure 3: **Spatial attention architectures.** *Top*: Illustration of different spatial attention architectures. Compared to prior methods, our proposed architecture ③ incorporates both global and local spatial attention and can be efficiently learned using a *patch-by-patch* strategy. *Bottom*: Long-term optical flows predicted with different spatial attention designs. We find that both global and local attention are crucial for improving tracking accuracy, as highlighted by the red circles. Additionally, our computationally efficient global attention design using *anchor tracks* (i.e., ③ W/o Local Attn) achieves similar accuracy to the more computationally-intensive CoTracker version ②.

*Computational complexity.* As shown in Fig. 3, applying spatial self-attention across all tokens in a frame (①) results in a computational cost of $TN^2$ making it impractical for dense tracking. To reduce complexity, CoTracker introduces virtual track tokens which are conceptually similar to learnable tokens introduced by DETR (Carion et al., 2020). It performs cross-attention back-and-forth between trajectory tokens and a small number of virtual tracks, and self-attention is only applied within the virtual tracks (②). This reduces the computational cost to $2TKN + TK^2 + KT^2$, assuming that the number of virtual tracks $K \ll N$. However, this reduction is still not enough for end-to-end tracking of every pixel in a high-resolution video. Although in practice, pixels could be divided into disjoint groups and perform attention within each group separately, this strategy is sub-optimal due to the interaction between tokens from different groups are ignored.

*Spatial granularity.* The use of virtual tracks assumes a small number of tokens and thus reduces spatial attention granularity, limiting the capacity to represent fine spatial details for dense tracking. Increasing virtual tracks improves accuracy but negates the complexity reduction.

**Method overview.** To further reduce computation cost without sacrificing accuracy, we adopt a strategy commonly used in optical flow (Teed & Deng, 2020): performing complex computations at the reduced spatial resolution, followed by lighter layers for upsampling. As shown in Fig. 2, we first run dense tracking on $1/r^2$ of the original resolution, reducing the computation cost by $1/r^2$. Then the reduced-resolution tracks are upsampled to full spatial resolution. In the following section, we use the notation $N = (H \times W)/r^2$ to denote the number of dense tracks at reduced resolution.

In Sec 3.2, we first discuss our design for reduced-resolution tracking with a new spatial attention architecture, which maintains linear complexity w.r.t. number of tracks while providing finer spatial granularity compared to CoTracker. More importantly, the new architecture can be learned end-to-end for pixel-wise dense tracking without test-train resolution discrepancies. Next in Sec 3.3 we introduce a new transformer-based upsampler that effectively predicts high-res tracking. Finally in Sec 3.4, we delve into details of depth representation that is crucial for accurate tracking in 3D.

### 3.2 JOINT GLOBAL AND LOCAL SPATIAL ATTENTION FOR EFFICIENT DENSE TRACKING

We design our global-local spatial attention mechanism based on three key criteria: (1) efficiency for both training and testing in dense settings, (2) ability to capture global motion across the image, and (3) ability to capture fine-grained motion details in the neighboring region for each track.

Figure 4: **Attention-based upsample module**. *Left*: We apply multiple blocks of local cross-attention to learn the upsampling weights for each pixel in the fine resolution. *Right*: The red circles highlight regions in the long-term flow maps where our attention-based upsampler produces more accurate predictions compared to RAFT's convolution-based upsampler.

**Global attention with sparse anchor tracks.** The spatial attention architecture from previous works, *i.e.* CoTracker (Karaev et al., 2023), which has linear complexity w.r.t. the number of tracks, can perform inference at reduced resolution. However, training remains computationally expensive, exceeding the 80GB GPU memory capacity even when tracking videos with a $96 \times 128$ reduced resolution. To improve training efficiency, we employ the following two strategies.

First, a patchwise training strategy is employed to reduce computation in training. At each iteration, we randomly crop small patches of size $h' \times w' = N'$ from the reduced resolution image, then perform dense tracking and obtain supervision within these patches. One issue of patchwise training is that it only computes spatial attention within a local patch without considering the rest region of the first frame. To mitigate this limitation, we augment the patch by introducing a sparse set of $M$ anchor tracks ($M \approx 10^2 \ll N$), with starting positions uniformly sampled across the first frame.

The second strategy involves computing virtual tracks by cross-attending only to the anchor tracks, instead of attending to all tracks as in CoTracker. As illustrated in ③ of Fig. 3, this approach reduces the cross-attention cost for computing virtual tracks from $TKN'$ to $TKM$. Consequently, the total cost for the new global attention becomes $TK(N' + 2M) + TK^2 + KT^2$, approximately halving the original cost of $2TKN' + TK^2 + KT^2$, assuming $T, K, M \ll N'$. As shown in Table 6b, this reduction in computation has minimal impact on tracking accuracy. An additional advantage of learning virtual tracks from anchor tracks is that the same set of anchor tracks is used during both patchwise training and testing on the full image, eliminating any train-test resolution mismatch.

**Dense local attention.** To capture fine-grained representations of local relations among dense tracks, we apply self-attention within very small spatial patches containing $L$ pixels, prior to cross-attending to the virtual tracks. Since self-attention is applied only within these small patches, this approach adds a marginal complexity of $O(TNL)$ during inference, where $L \ll N$ is the number of tracks per patch. Experiments (Table 6b) show that incorporating dense local attention significantly improves dense tracking accuracy.

*In summary*, our joint Global-Local spatial attention has roughly the same computational cost as CoTracker's global-only attention but (i) captures both global motion and fined-grained spatial relations, (ii) enables end-to-end training, and (iii) achieves significantly better performance in dense tracking (see Table 2, Table 3 and the qualitative results in our supplementary).

### 3.3 HIGH-RESOLUTION TRACK UPSAMPLER

Given the dense tracks extracted at a reduced spatial resolution of $\frac{H}{r} \times \frac{W}{r}$, the next step is to upsample them to the full resolution $H \times W$. In the context of optical flow, a common upsampling approach expresses the flow for each fine-resolution pixel as a convex combination of its nearest neighbor flows estimated in a coarse resolution (Teed & Deng, 2020). The weights for the combination are learned via a convnet. In contrast, we propose an attention-based upsampling mechanism that more effectively captures the correlation between each fine-resolution pixel and its neighbors at the coarse resolution. We demonstrate the efficacy in our experiments (see Figure 4 and Table 6c).

Starting from the first input frame of the window (for clarity, we omit the frame subscript $t$ in this section), the frame is initially processed through a lightweight convolutional backbone

to extract a coarse resolution feature map, $\mathcal{F}_{coarse} \in \mathbb{R}^{\frac{H}{r} \times \frac{W}{r} \times D}$, where $D$ is the channel dimension. This coarse feature map is then upsampled using a convolutional decoder to produce a fine-resolution feature map, $\mathcal{F}_{fine} \in \mathbb{R}^{H \times W \times D}$. Each fine-resolution pixel $(u, v)$, with the feature vector $\mathcal{F}_{fine}^{(u,v)} \in \mathbb{R}^{1 \times D}$, cross-attends to a $\kappa \times \kappa$ neighborhood centered on its corresponding coarse location $(u', v')$ in the coarse resolution, where $u' = u/r$ and $v' = v/r$, using subpixel accuracy for $u'$ and $v'$. The neighboring coarse features is defined as the set $\{\mathcal{F}_{coarse}^{(u'_j, v'_j)}\}_{j=1}^{\kappa \times \kappa} \in \mathbb{R}^{(\kappa \times \kappa) \times D}$. Specifically, the fine-resolution feature map is extracted through a cross-attention operation:

$$\mathcal{F}_{fine}^{(u,v)} = \boldsymbol{a}(u, v) \cdot \boldsymbol{v}(\{\mathcal{F}_{coarse}^{(u'_j, v'_j)}\}_{j=1}^{\kappa \times \kappa}) \tag{3}$$

where the cross-attention scores $\boldsymbol{a}(u, v)$ are computed as:

$$\boldsymbol{a}(u, v) = softmax\Big(\boldsymbol{q}(\mathcal{F}_{fine}^{(u,v)}) \cdot \boldsymbol{k}(\{\mathcal{F}_{coarse}^{(u'_j, v'_j)}\}_{j=1}^{\kappa \times \kappa}) + m \cdot ||(u', v') - \{u'_j, v'_j\}_{j=1}^{\kappa \times \kappa}||_1\Big) \tag{4}$$

and $\boldsymbol{q}(\cdot), \boldsymbol{k}(\cdot), \boldsymbol{v}(\cdot)$ are linear transformations for the queries, keys, and values respectively. The term added to the above dot product represents a static, non-learned spatial bias inspired by Alibi (Press et al., 2022). In our case, we bias the query-key attention scores between pixels with a penalty proportional to their distance between their positions (L1 distance in our implementation). We refine the fine-resolution feature map by applying a series of $\tau$ multi-head, Alibi-modified cross-attention blocks, as described in Eq. 3. Finally, we use a MLP to predict the weight map $\mathcal{W} = MLP(\mathcal{F}_{fine})$, where $\mathcal{W} \in \mathbb{R}^{H \times W \times (\kappa \times \kappa)}$. This allows us to compute the high-resolution tracking by taking a weighted average of the coarse-resolution tracks using the predicted weight map. We found that more temporally consistent results are produced when the weights are estimated once for the first frame of the time window, and then the same weights are reused for the rest of the frames.

## 3.4 DELVING DEEPER INTO DEPTH REPRESENTATION

Prior works in 3D tracking have primarily focused on exploring different designs of 3D features, such as depth correlation features (Wang et al., 2024a) and triplane features (Xiao et al., 2024). Our experiments reveal that the choice of **depth representation**, which is a previously overlooked factor, has a much more significant impact. In previous works, 3D features were typically computed in Euclidean space, with depth normalized to a fixed range, and the network predicted the difference in normalized depth. We find that alternative depth representations, such as inverse depth $1/d$ and log depth $\log(d)$, improve 3D accuracy, with log depth offering the greatest boost (Table 6a).

This improvement can be intuitively explained: Euclidean depth evenly distributes granularity along the depth axis, which is suboptimal since objects of interest are typically closer. Inverse or log depth enhances precision for nearby regions, where visual-based depth estimation methods tend to be more reliable, while tolerating higher uncertainty for distant areas. This reasoning also underlies why monocular depth estimation methods are often trained to output either inverse depth or log depth (Eigen et al., 2014; Wang et al., 2019; Ranftl et al., 2022).

More critically, switching the network output from $\Delta d_t = d_t - d_1$ to $\Delta \log(d_t) = \log(d_t) - \log(d_1) = \log(d_t/d_1)$, and similarly adjusting the depth correlation feature to a log depth correlation feature, improves robustness to imperfections in input depth maps. The *depth change ratio* $d_t/d_1$, being scale-invariant, effectively decouples the network from the arbitrary scale of the input depth maps. This ratio also aligns with the concept of optical expansion (Swanston & Gogel, 1986; Schrater et al., 2001), where objects appear larger as they approach the camera. Thus, estimating depth change ratios directly from visual features makes the network less dependent on depth map accuracy, a strategy similarly used in scene flow estimation (Yang & Ramanan, 2020).

## 4 EXPERIMENTS

### 4.1 IMPLEMENTATION DETAILS

**Training data.** We leverage the Kubric simulator (Greff et al., 2022) to generate 5,632 training RGB-D videos and 143 testing videos, featuring falling rigid objects against diverse backgrounds. Dense trajectories are annotated for every pixel in the first, middle, and last frames of each video. To augment the training set, we apply random geometric and color augmentations to the images and introduce noise to the depth maps.

**Training loss.** We supervise the model using both the low-res and the upsampled predictions. The total loss is defined as $\lambda_{2d}\mathcal{L}_{2D} + \lambda_{depth}\mathcal{L}_{depth} + \lambda_{visib}\mathcal{L}_{visib}$, where $\mathcal{L}_{2D}$ and $\mathcal{L}_{depth}$ are

| Methods | CVO-Clean (7 frames) | | CVO-Final (7 frames) | | CVO-Extended (48 frames) | |
|---|---|---|---|---|---|---|
| | EPE↓ (*all/vis/occ*) | IoU↑ | EPE↓ (*all/vis/occ*) | IoU↑ | EPE↓ (*all/vis/occ*) | IoU↑ |
| RAFT (Teed & Deng, 2020) | 2.48 / 1.40 / 7.42 | 57.6 | 2.63 / 1.57 / 7.50 | 56.7 | 21.80 / 15.4 / 33.4 | 65.0 |
| MFT (Neoral et al., 2024) | 2.91 / 1.39 / 9.93 | 19.4 | 3.16 / 1.56 / 10.3 | 19.5 | 21.40 / 9.20 / 41.8 | 37.6 |
| TAPIR (Doersch et al., 2023) | 3.80 / 1.49 / 14.7 | 73.5 | 4.19 / 1.86 / 15.3 | 72.4 | 19.8 / 4.74 / 42.5 | 68.4 |
| CoTracker (Karaev et al., 2023) | 1.51 / 0.88 / 4.57 | 75.5 | 1.52 / 0.93 / 4.38 | 75.3 | 5.20 / 3.84 / 7.70 | 70.4 |
| DOT (Le Moing et al., 2024) | 1.29 / 0.72 / 4.03 | **80.4** | 1.34 / 0.80 / 3.99 | **80.4** | 4.98 / 3.59 / 7.17 | 71.1 |
| SceneTracker (Wang et al., 2024a) | 4.40 / 3.44 / 9.47 | - | 4.61 / 3.70 / 9.62 | - | 11.5 / 8.49 / 17.0 | - |
| SpatialTracker (Xiao et al., 2024) | 1.84 / 1.32 / 4.72 | 68.5 | 1.88 / 1.37 / 4.68 | 68.1 | 5.53 / 4.18 / 8.68 | 66.6 |
| DOT-3D | 1.33 / 0.75 / 4.16 | 79.0 | 1.38 / 0.83 / 4.10 | 78.8 | 5.20 / 3.58 / 7.95 | 70.9 |
| Ours (2D) | **0.89 / 0.46 / 2.96** | 78.3 | **0.97 / 0.55 / 2.96** | 77.7 | **3.63** / 2.67 / **5.24** | **71.6** |
| Ours (3D) | 0.94 / 0.51 / 2.97 | 78.7 | 1.03 / 0.61 / 3.03 | 78.3 | 3.67 / **2.64** / 5.30 | 70.1 |

Table 2: **Long-range optical flow results** on CVO (Wu et al., 2023; Le Moing et al., 2024).

| Methods | Kubric-3D (24 frames) | | | Time |
|---|---|---|---|---|
| | AJ↑ | APD$_{3D}$ ↑ | OA↑ | |
| SpatialTracker | 42.7 | 51.6 | 96.5 | 9mins |
| SceneTracker | - | 65.5 | - | 5mins |
| DOT-3D | 72.3 | 77.5 | 88.7 | **0.15mins** |
| Ours | **81.4** | **88.6** | **96.6** | 0.5mins |

Table 3: **Dense 3D tracking results** on the Kubric3D dataset.

| Methods | LSFOdyssey | | |
|---|---|---|---|
| | AJ↑ | APD$_{3D}$ ↑ | OA↑ |
| SpatialTracker | 5.7 | 9.9 | 84.0 |
| SceneTracker[‡] | - | 57.7 | - |
| Ours | 29.4 | 39.6 | **84.4** |
| Ours[‡] | **50.1** | **69.7** | 83.9 |

Table 4: **3D tracking results** on the LSFOdyssey benchmark. [‡] denotes models trained with LS-FOdyssey training set.

the L1 losses comparing the predicted 2D coordinates and inverse depth with the ground truth, and $\mathcal{L}_{visib}$ is the binary cross entropy loss for visibility prediction. We empirically set weightings $\lambda_{2d}, \lambda_{depth}, \lambda_{visib}$ to be $100.0, 1.0, 0.1$.

**Training details.** Training details are included in the appendix.

## 4.2 COMPARISON TO PRIOR WORKS

**Baselines.** We evaluate our method against prior optical flow and point tracking methods. Particularly, we perform a close comparison against DOT (Le Moing et al., 2024), a recent SoTA method designed for dense 2D tracking. We implemented a 3D extension of DOT, named DOT-3D, where we incorporate depth map input into its optical flow module and add a head to output $\log(d_t/d_1)$.

**Benchmark datasets.** We evaluate the performance of our approach across multiple tracking scenarios, including long-range 2D optical flow, dense 3D tracking, and 3D point tracking benchmarks.
• *Long-range 2D optical flow:* We use the **CVO** (Wu et al., 2023) test set, which originally includes two subsets: *CVO-Clean* and *CVO-Final*, the latter incorporating motion blur. Each split contains approximately 500 videos with 7 frames captured at 60 FPS. Following the comparison in DOT(Le Moing et al., 2024), we introduce an additional split, *CVO-Extended*, which includes 500 videos of 48 frames rendered at 24 FPS. All videos in the CVO dataset are annotated with dense, long-range 2D optical flow and occlusion masks.
• *Dense 3D tracking:* We use our generated **Kubric** test split with 143 videos, each with 24 frames.
• *3D point tracking:* We use two benchmarks: (1) **TAP-Vid3D** includes videos from 3 datasets with different scenarios: DriveTrack (Balasingam et al., 2024), PStudio (Joo et al., 2017), and Aria (Pan et al., 2023) with total 4569 videos for evaluation, where the number of frames varies from 25 to 300 per video. (2) **LSFOdyssey** contains 90 40-frame videos derived from the PointOdyssey dataset (Zheng et al., 2023). Both datasets provide sparse trajectories and occlusion annotations.

**Metrics.** For *long-range optical flow* benchmark, we follow Le Moing et al. (2024) and report the end-point-error (EPE) between the predicted flows and groundtruth flows for both *visible*, *occluded* points and the intersection over union (IoU) between predicted and ground-truth occluded regions in visibility masks. For *dense 3D tracking* and *3D point tracking* benchmarks, we follow Koppula et al. (2024) and report APD$_{3D}$ ($< \delta_{avg}$) which measures the average percent of points within $\delta_x$ error threshold, occlusion accuracy OA measures the accuracy of visibility prediction and the average Jaccard (AJ) which evaluates both occlusion and position accuracy.

| Methods | Aria | | | DriveTrack | | | PStudio | | | Average | | |
|---|---|---|---|---|---|---|---|---|---|---|---|---|
| | AJ↑ | APD$_{3D}$↑ | OA↑ | AJ↑ | APD$_{3D}$↑ | OA↑ | AJ↑ | APD$_{3D}$↑ | OA↑ | AJ↑ | APD$_{3D}$↑ | OA↑ |
| TAPIR[†] + COLMAP | 7.1 | 11.9 | 72.6 | 8.9 | 14.7 | 80.4 | 6.1 | 10.7 | 75.2 | 7.4 | 12.4 | 76.1 |
| CoTracker[†] + COLMAP | 8.0 | 12.3 | 78.6 | 11.7 | 19.1 | 81.7 | 8.1 | 13.5 | 77.2 | 9.3 | 15.0 | 79.1 |
| BootsTAPIR[†] + COLMAP | 9.1 | 14.5 | 78.6 | 11.8 | 18.6 | 83.8 | 6.9 | 11.6 | **81.8** | 9.3 | 14.9 | 81.4 |
| CoTracker[†] + UniDepth | 13.0 | 20.9 | 84.9 | 12.5 | 19.9 | 80.1 | 6.2 | 13.5 | 67.8 | 10.6 | 18.1 | 77.6 |
| TAPTR[†] + UniDepth | 15.7 | 24.2 | 87.8 | 12.4 | 19.1 | 84.8 | 7.3 | 13.5 | 84.3 | 11.8 | 18.9 | **85.6** |
| LocoTrack[†] + UniDepth | 15.1 | 24.0 | 83.5 | 13.0 | 19.8 | 82.8 | 7.2 | 13.1 | 80.1 | 11.8 | 19.0 | 82.3 |
| SpatialTracker + UniDepth | 13.6 | 20.9 | **90.5** | 8.3 | 14.5 | 82.8 | 8.0 | 15.0 | 75.8 | 10.0 | 16.8 | 83.0 |
| SceneTracker + UniDepth | - | 23.1 | - | - | 6.8 | - | - | 12.7 | - | - | 14.2 | - |
| DOT-3D + UniDepth | 13.8 | 22.1 | 85.5 | 11.8 | 17.9 | 82.3 | 3.2 | 5.3 | 52.5 | 9.6 | 15.1 | 73.4 |
| Ours + UniDepth | **16.6** | **24.4** | 86.8 | **14.6** | **22.5** | **85.8** | **8.2** | **15.0** | 76.4 | **13.1** | **20.6** | 83.0 |

Table 5: **3D tracking results** on the TAP-Vid3D Benchmark. We report the 3D average jaccard (AJ), average 3D position accuracy (APD$_{3D}$), and occlusion accuracy (OA) across datasets Aria, DriveTrack, and PStudio using UniDepth and ZoeDepth for depth estimation.[†] denotes using depth to lift 2D tracks to 3D tracks. We re-evaluated SpatialTracker and SceneTracker using their publicly available code and checkpoints, following the same inference procedure as our method. We note that the results differ slightly from the numbers reported in the TAP-Vid3D paper.

| Depth Repr. | Network Output | TAP-Vid3D (*Avg.*) | |
|---|---|---|---|
| | | AJ↑ | APD$_{3D}$↑ |
| $d$ | $d_t - d_1$ | 9.0 | 15.0 |
| $1/d$ | $1/d_t - 1/d_1$ | 9.4 | 15.6 |
| $\log(d)$ | $\log(d_t/d_1)$ | **13.1** | **20.6** |

(a) Depth representation

| Global Attn. | Local Attn. | CVO (*Extended*) | |
|---|---|---|---|
| | | EPE↓ | OA↑ |
| ✗ | ✗ | 10.0 / 4.84 / 18.1 | 65.7 |
| ✗ | ✓ | 8.01 / 3.89 / 13.91 | 69.0 |
| ② CoTracker | ✗ | 3.72 / 2.78 / 5.44 | 70.1 |
| ③ Ours | ✗ | 3.73/ 2.78 / 5.47 | 70.0 |
| ③ Ours | ✓ | **3.67 / 2.64 / 5.30** | **70.1** |

(b) Spatial attention design

| Upsample Method | CVO (*Extended*) | |
|---|---|---|
| | EPE ↓ | OA ↑ |
| Bilinear | 5.31 / 4.14 / 7.94 | 68.9 |
| NN | 5.34 / 4.17 / 7.98 | 66.9 |
| 3D KNN | 4.59 / 3.41 / 7.07 | 68.9 |
| ConvUp | 4.27 / 3.09 / 6.73 | 70.2 |
| AttentionUp | 3.73 / 2.73 / 5.35 | **70.3** |
| AttentionUp + Alibi | **3.67 / 2.64 / 5.30** | 70.1 |

(c) Upsampler design

Table 6: **Ablation studies** (a) different depth representations on TAP-Vid3D (b) different spatial attention designs on the CVO (Extended) (c) different upsampler designs on CVO (*Extended*).

**Long-range optical flow results.** We first compare our method against baseline approaches on the dense 2D tracking task (see Tab. 2). This experiment isolates the evaluation from additional 3D features and supervision, making it a straightforward assessment of our proposed network architecture for handling dense per-pixel tracking. We find that our method significantly outperforms all previous approaches, including the recent SOTA method, DOT, in terms of positional accuracy. The improvement is particularly noticeable when visualizing the results (see Appendix), where the trajectories predicted by DOT tend to become unstable once the tracked pixel is occluded or moves out of view. This highlights the importance of maintaining temporal attention in the tracking network, a feature absent in DOT.

Additionally, we compare both DOT and our method with and without 3D supervision. We find that the variants are nearly equivalent, although the quantitative performance slightly decreases for the 3D-supervised versions. We also observe that our visibility mask accuracy is on par with CoTracker, from which our method is derived, though it is marginally lower than DOT. These discrepancies could potentially be addressed by adjusting the weightings of different terms in the training loss.

**Dense 3D Tracking results.** We report the results of dense 3D tracking on the Kubric synthetic test set, where our approach significantly outperforms other methods in both accuracy and runtime. We visualize of 3D dense tracking results in figure 5. Compared to SceneTracker and SpatialTracker, our method excels at accurately predicting the locations of moving objects and preserving object shapes throughout the video. Please find more qualitative results in the supplementary.

**3D point tracking benchmarks results.** To further evaluate the generalizability of our approach on in-the-wild videos, we assess its performance on the TAPVid-3D dataset using depth maps estimated by either UniDepth (Piccinelli et al., 2024) or ZoeDepth (Bhat et al., 2023). The results, summarized in Table 5, show that our method consistently outperforms previous approaches, including SpatialTracker, SceneTracker, and 3D-lifted versions of state-of-the-art 2D tracking methods. Our approach demonstrates improvements across most sub-datasets, as well as in the overall average.

We also evaluate our approach on the LSFOdyssey dataset (Wang et al., 2024a), as shown in Tab. 4. In this benchmark, SceneTracker, trained specifically on the same domain, outperforms both Spa-

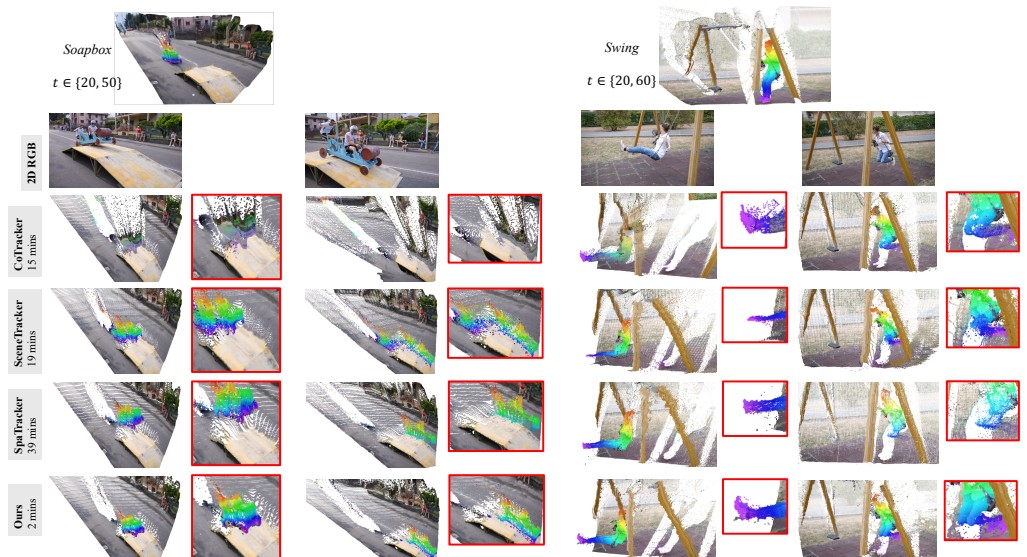

Figure 5: **Qualitative results of dense 3D tracking on in-the-wild videos** between CoTracker + UniDepth, SceneTracker, SpatialTracker and our method. We densely track every pixel from the first frame of the video in 3D space, the moving objects are highlighted as rainbow color. Our method accurately tracks the motion of foreground objects while maintaining stable backgrounds.

tialTracker and our model, which were trained on the Kubric dataset. To ensure a fair comparison, we fine-tuned our model for just one epoch on the LSFOdyssey training set and observed substantial performance improvements, surpassing SceneTracker.

### 4.3 ABLATION STUDY

**Study on the 3D representation** is presented in Tab. 6a. We find that representing depth using log depth significantly improves 3D tracking accuracy compared to using depth and inverse depth.

**Study on the design of spatial attention** is shown in Tab. 6b. We evaluated different spatial attention variants, as illustrated in Fig.3, comparing approaches with and without global or local attention. We also compared two versions of global attention: the one used in CoTracker, which cross-attends virtual tracks back and forth (illustrated in ② of Fig.3), and our proposed method of cross-attending virtual tracks with anchor tracks (illustrated in ③ of Fig. 3). Our results show that both global and local attention are crucial, and our design of global attention achieves comparable accuracy to CoTracker while being more computationally efficient.

**Study on the design of upsampler** is reported in Tab. 6c. We compared our approach against upsampling methods using non-learnable operators (bilinear, nearest neighbor, and 3D K-nearest neighbor) and the CNN-based upsampler from RAFT (Teed & Deng, 2020). Our method noticeably outperforms all of these approaches.

## 5 CONCLUSION

We presented a method that efficiently tracks every pixel of a frame throughout a video, demonstrating state-of-the-art accuracy in dense 2D/3D tracking while running significantly faster than existing 3D tracking methods. Despite these successes, our method shares some common **limitations** with previous point-tracking approaches due to its relatively short temporal processing windows. It may fail to track points that remain occluded for extended periods, and it currently performs best with videos of fewer than a few hundred frames. Additionally, our 3D tracking performance is closely tied to the accuracy and temporal consistency of the off-the-shelf monocular depth estimation. We anticipate that our method will benefit from recent rapid advancements in monocular depth estimation research (Hu et al., 2024).

**Acknowledgements** Evangelos Kalogerakis has received funding from the European Research Council (ERC) under the Horizon research and innovation programme (Grant agreement No. 101124742).

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

| Methods | Aria AJ↑ | APD$_{3D}$↑ | OA↑ | DriveTrack AJ↑ | APD$_{3D}$↑ | OA↑ | PStudio AJ↑ | APD$_{3D}$↑ | OA↑ | Average AJ↑ | APD$_{3D}$↑ | OA↑ |
|---|---|---|---|---|---|---|---|---|---|---|---|---|
| TAPIR† + ZoeDepth | 9.0 | 14.3 | 79.7 | 5.2 | 8.8 | 81.6 | 10.7 | 18.2 | 78.7 | 8.3 | 13.8 | 80.0 |
| CoTracker† + ZoeDepth | 10.0 | 15.9 | 87.8 | 5.0 | 9.1 | 82.6 | 11.2 | 19.4 | 80.0 | 8.7 | 14.8 | 83.4 |
| BootsTAPIR† + ZoeDepth | 9.9 | 16.3 | 86.5 | 5.4 | 9.2 | 85.3 | 11.3 | 19.0 | 82.7 | 8.8 | 14.8 | 84.8 |
| TAPTR† + ZoeDepth | 9.1 | 15.3 | 87.8 | 7.4 | 12.4 | 84.8 | 10.0 | 17.8 | 84.3 | 8.8 | 15.2 | 85.6 |
| LocoTrack† + ZoeDepth | 8.9 | 15.1 | 83.5 | 7.5 | 12.3 | 82.8 | 9.7 | 17.1 | 80.1 | 8.7 | 14.8 | 82.1 |
| SpatialTracker + ZoeDepth | 9.2 | 15.1 | 89.9 | 5.8 | 10.2 | 82.0 | 9.8 | 17.7 | 78.0 | 8.3 | 14.3 | 83.3 |
| SceneTracker + ZoeDepth | - | 15.1 | - | - | 5.6 | - | - | 16.3 | - | - | 12.3 | - |
| Ours + ZoeDepth | 10.1 | 16.2 | 84.7 | 7.8 | 12.8 | 87.2 | 10.2 | 17.8 | 74.5 | 9.4 | 15.6 | 82.1 |

Table 7: **Additional 3D tracking results** on the TAP-Vid3D Benchmark. We report the 3D average jaccard (AJ), average 3D position accuracy (APD$_{3D}$), and occlusion accuracy (OA) across datasets Aria, DriveTrack, and PStudio using ZoeDepth for depth estimation.† denotes using depth to lift 2D tracks to 3D tracks.

| Methods | Kinetics AJ↑ | APD$_{2D}$↑ | OA↑ | DAVIS AJ↑ | APD$_{2D}$↑ | OA↑ | RGB-Stacking AJ↑ | APD$_{2D}$↑ | OA↑ |
|---|---|---|---|---|---|---|---|---|---|
| TAP-Net (Doersch et al., 2022) | 38.5 | 54.4 | 80.6 | 33.0 | 48.6 | 78.8 | 54.6 | 68.3 | 87.7 |
| MFT (Neoral et al., 2024) | 39.6 | 60.4 | 72.7 | 47.3 | 66.8 | 77.8 | - | - | - |
| PIPs (Harley et al., 2022) | 31.7 | 53.7 | 72.9 | 42.2 | 64.8 | 77.7 | 15.7 | 28.4 | 77.1 |
| OmniMotion (Wang et al., 2023a) | - | - | - | 46.4 | 62.7 | 85.3 | 69.5 | 82.5 | 90.3 |
| TAPIR (Doersch et al., 2023) | 49.6 | 64.2 | 85.0 | 56.2 | 70.0 | 86.5 | 54.2 | 69.8 | 84.4 |
| CoTracker (Karaev et al., 2023) | 48.7 | 64.3 | 86.5 | 60.6 | 75.4 | 89.3 | 63.1 | 77.0 | 87.8 |
| DOT (Le Moing et al., 2024) | 48.4 | 63.8 | 85.2 | 60.1 | 74.5 | 89.0 | 77.1 | 87.7 | 93.3 |
| BootsTAPIR (Doersch et al., 2024) | 54.6 | 68.4 | 86.5 | 61.4 | 73.6 | 88.7 | 70.8 | 83.0 | 89.9 |
| TAPTR (Li et al., 2024b) | 49.0 | 64.4 | 85.2 | 63.0 | 76.1 | 91.1 | - | - | - |
| TAPTRv2 (Li et al., 2024a) | 49.7 | 64.2 | 85.7 | 63.5 | 75.9 | 91.4 | - | - | - |
| LocoTrack (Cho et al., 2024b) | 52.9 | 66.8 | 85.3 | 63.0 | 75.3 | 87.2 | 69.0 | 83.2 | 89.5 |
| SpatialTracker (Xiao et al., 2024) | 50.1 | 65.9 | 86.9 | 61.1 | 76.3 | 89.5 | 63.5 | 77.6 | 88.2 |
| SceneTracker (Wang et al., 2024a) | - | 66.5 | - | - | 71.8 | - | - | 73.3 | - |
| DOT-3D | 48.1 | 63.7 | 85.9 | 61.2 | 75.3 | 88.1 | 76.3 | 86.6 | 92.1 |
| Ours (2D) | 50.3 | 63.5 | 83.2 | 64.2 | 77.3 | 87.8 | 73.4 | 82.4 | 89.6 |
| Ours | 49.5 | 63.3 | 82.2 | 62.7 | 76.7 | 88.2 | 74.2 | 83.5 | 90.0 |

Table 8: **2D Tracking Results** on the TAP-Vid Benchmark (Doersch et al., 2022) (*query-first* mode). We report the average jaccard (AJ), average 2D position accuracy (APD$_{2D}$), and occlusion accuracy (OA) on the Kinetics (Carreira & Zisserman, 2017), DAVIS (Pont-Tuset et al., 2017) and RGB-Stacking (Lee et al., 2021) datasets.

# A  APPENDIX

## A.1  IMPLEMENTATION DETAILS

**Implementation details.** We use the same backbone as Karaev et al. (2023); Harley et al. (2022), which consists of 6 residual blocks, outputting feature maps with a dimension of 256. Unlike Co-Tracker, which extracts a single-scale feature map and applies pooling later in the correlation module, we directly generate a pyramid of feature maps at scales 2, 4, and 8. We perform dense pixel tracking at a resolution of $H/4 \times W/4$ (with $r = 4$). The transformer network $\Phi$ is composed of 6 spatial and temporal attention blocks, utilizing 8 attention heads and 384 hidden channels. The number iteration step is set to 6. The number of anchor tracks is set to $9 \times 12$ during training. In the patch-wise dense local attention, we use a patch size of 6, resulting in $L = 36$ tracks per patch. In the high-resolution track upsampler, we use 9 neighbors (with $\kappa = 3$) and 2 cross-attention blocks.

**Training details.** We first pretrain the model with 2D loss and visibility loss for 100k iterations, then train with the full loss for another 100k iterations. All stages are conducted on a machine with 8 A100 GPUs. We use the AdamW optimizer and the batch size is set to 1 for each GPU. The learning

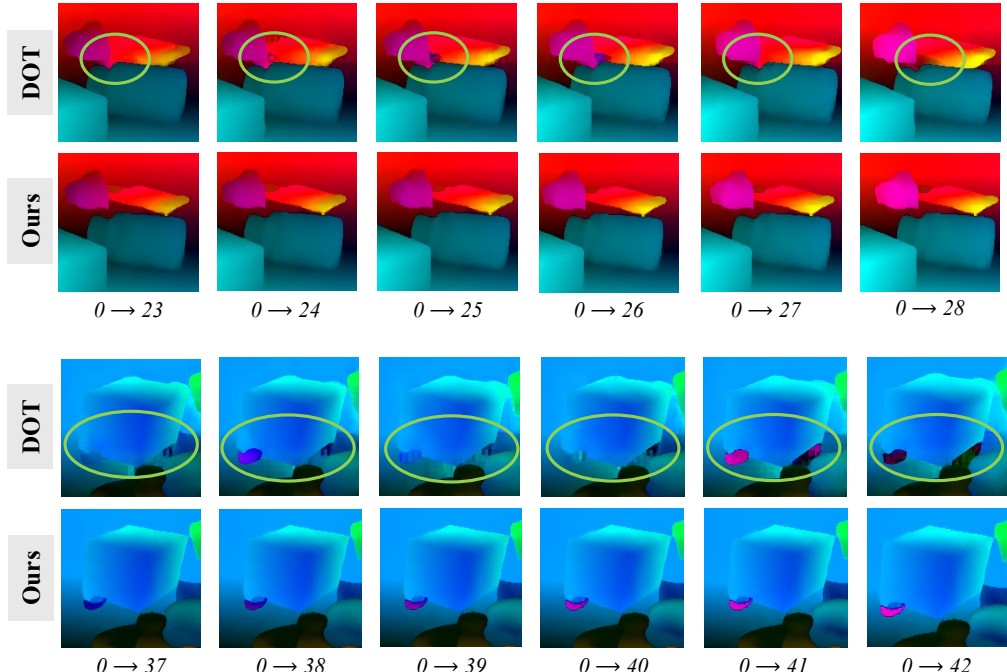

Figure 6: **Comparison of long-range optical flow predictions**: We predict optical flows from the first frame to subsequent frames of the video. DOT (Le Moing et al., 2024), which lacks strong temporal correlation, suffers from a noticeable "flickering" effect (green circle), particularly at the boundaries between foreground and background objects. In contrast, our method ensures a smooth and consistent transition over time, effectively reducing artifacts at object boundaries.

rate is initialized to $10^{-4}$ and scheduled by a linear one cycle (Smith & Topin, 2019). During training, to save the GPU memory consumption, we randomly sample a patch of size $N' = 30 \times 40$ from the dense 3D trajectory map as supervision. The input video is resized to $384 \times 512$ in both training and testing. For in-the-wild video, we leverage ZoeDepth (Bhat et al., 2023) and UniDepth (Piccinelli et al., 2024) to obtain video depth.

### A.2   3D POINT TRACKING

We additionally report the performance on the TAPVid-3D dataset using depth maps estimated by ZoeDepth (Bhat et al., 2023) in Table 7.

### A.3   2D POINT TRACKING

**Sparse Tracking Setting.** In the sparse tracking scenario, where there are fewer than 10K points and they are sparsely distributed across the image, our model can seamlessly switch to a sparse mode by disabling the local attention in the Transformer and the Upsampler. This adjustment is made without requiring any changes to the overall architecture. This mode enables efficient evaluation on sparse tracking benchmarks such as TAP-Vid (Doersch et al., 2022) and TAP-Vid3D (Koppula et al., 2024).

**2D point tracking benchmark results.** We also evaluate the performance of our method on 2D point tracking on the TAP-Vid dataset, containing videos from 3 datasets: DAVIS (Pont-Tuset et al., 2017) with 30 in-the-wild videos, RGB-Stacking (Carreira & Zisserman, 2017) with 50 synthetic sequences, and Kinetics (Lee et al., 2021) with 1144 real videos. The results are reported in Table 8. Our approach outperforms 3D tracking methods, including SpatialTracker (Xiao et al., 2024), SceneTracker (Wang et al., 2024a), and DOT-3D, in most of the sub-datasets while having competitive performance with other SoTA approaches designed for sparse 2D tracking only.

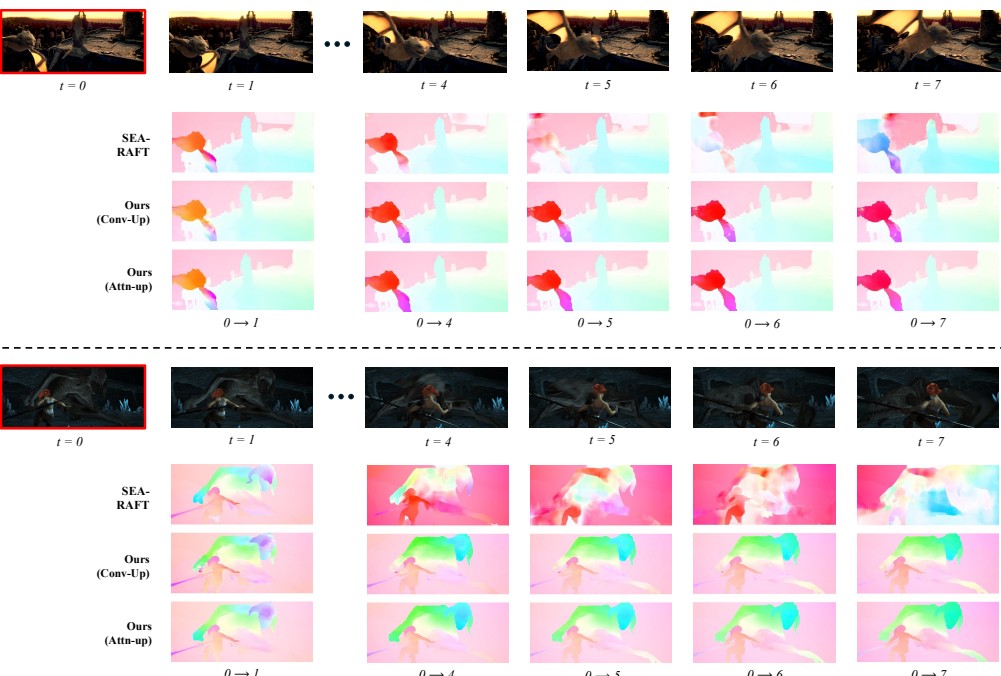

Figure 7: **Comparison of long-range optical flow predictions on Sintel dataset** between our approach (with CNN-based/attention-based upsampler) and SEA-RAFT. We predict optical flows from the first frame to subsequent frames of the video. While SEA-RAFT excels at short-range flow prediction (first column), it fails to predict flow between far-away frames. In contrast, our approach, with an attention-based upsampler, achieves smooth and consistent predictions across frames, outperforming the baseline using convolutional upsampler.

|  | **CVO** (*Extended*) | |
|---|---|---|
|  | EPE ↓ | OA ↑ |
| W/o anchor tracks | 4.50 / 3.10 / 6.95 | 69.7 |
| With anchor tracks | **3.63 / 2.67 / 5.24** | **71.6** |

Table 9: Ablation on the role of *anchor tracks*.

### A.4 MORE RESULTS

**Qualitative results of long-range 2D flows** are visualized in figure 6. Thanks to the strong temporal consistency of our method, it produces smooth predictions over time, effectively avoiding the 'flicker' artifacts that are commonly observed in per-frame optical flow predictions, such as those produced by DOT(Le Moing et al., 2024).

**Qualitative comparison of the design of upsampler** on the Sintel dataset (Butler et al., 2012) is visualized in figure 7. Our attention-based upsampler (last row) significantly outperforms the CNN-based upsampler from RAFT (Teed & Deng, 2020) (second row), with better detail and less artifacts. For reference, we include predictions from the state-of-the-art optical flow model SEA-RAFT (Wang et al., 2025) (first row), which, despite being trained on Sintel, excels at short-range predictions (first column) but fails with long-range flows.

We also compare our long-range flow predictions between SEA-RAFT, DOT, and our method on in-the-wild videos. The results are visualized in figure 8 showing that our approach achieves better temporal smoothness and less artifact for long-range motion prediction.

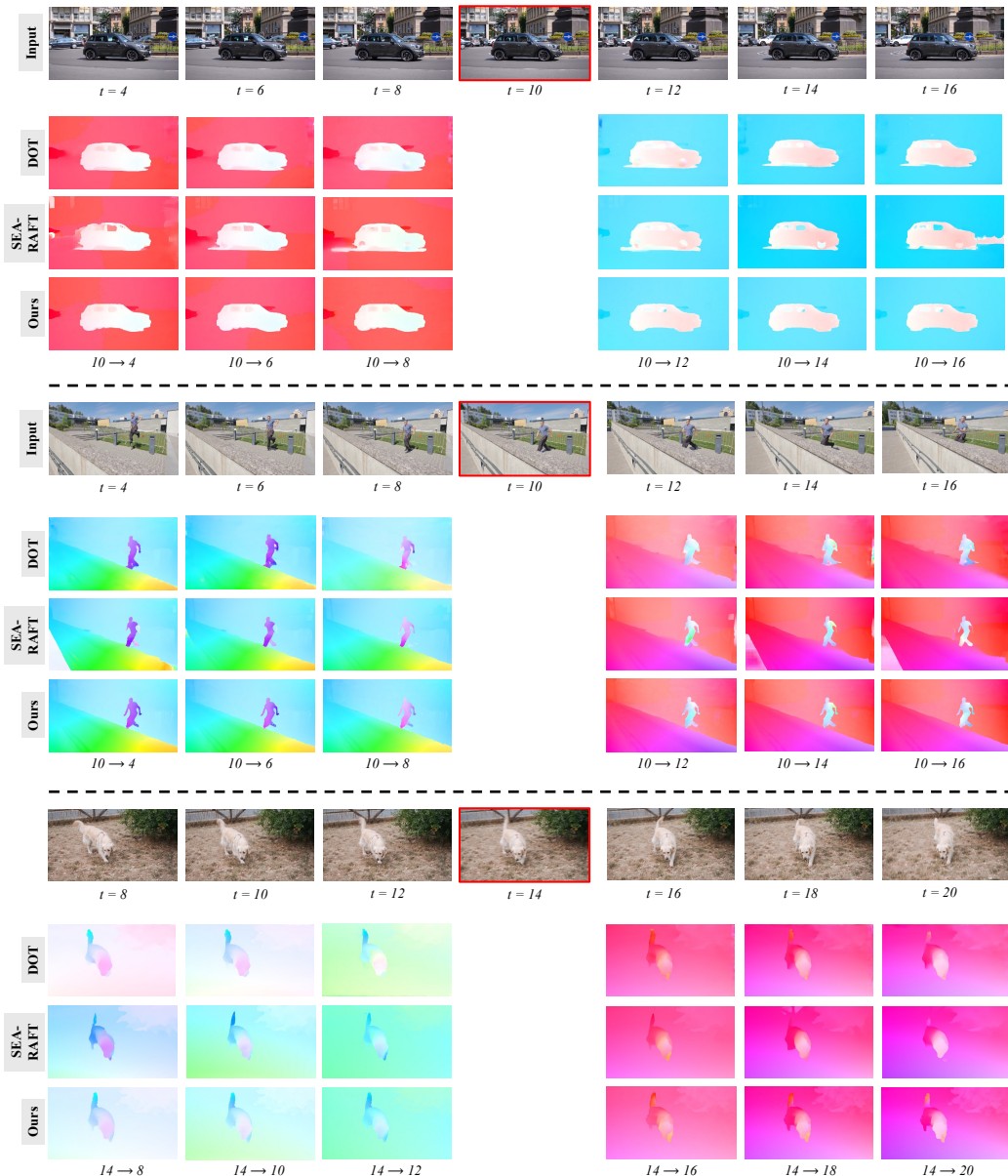

Figure 8: **Comparison of long-range optical flow predictions on in-the-wild videos** between DOT, SEA-RAFT, and our approach.

**Ablation on the anchor tracks** is shown in Table 9. Removing the anchor tracks during training increases the EPE by nearly 1.0, meaning that *the anchor tracks are important to avoid train-test mismatch*.

## A.5 FURTHER DISCUSSIONS ON THE 3D TRACKING PERFORMANCE

As shown in Table 5 of the main paper, baseline methods (2D tracking + depth estimation) significantly underperform compared to our end-to-end 3D tracking approach. Interestingly, better 2D tracking models do not always lead to better 3D tracking performance (e.g., BootsTAPIR *versus* CoTracker). This is because the performance of these baselines heavily relies on the quality and scale-consistency of the input video depth. However, frame-wise depth estimators like ZoeDepth and UniDepth lack scale-consistency (see our webpage), where even small depth errors can cause

significant 3D location discrepancies.

Furthermore, 2D tracking + depth estimation can only predict motion for visible points within the viewpoint, restricting applications such as 3D/4D reconstruction, which require complete motion predictions, including occluded and out-of-frame regions.

These challenges underline the need for end-to-end 3D tracking models to achieve reliable 3D performance. Our approach addresses these limitations and is also complementary to recent advances in 2D point tracking, enabling us to integrate their findings to further enhance both 2D and 3D tracking performance.

### A.6 APPLICATION

**Dynamic video pose estimation.** DUSt3R (Wang et al., 2024b) introduced a paradigm for estimating static 3D scene geometry and camera poses from image sets. By training a model on large-scale data, it predicts aligned 3D point maps for image pairs, followed by lightweight optimization to obtain scale-consistent depth maps and camera poses. However, this approach fails on videos with dynamic objects. MonST3R (Zhang et al., 2024) extended DUSt3R to dynamic videos by fine-tuning the model on dynamic datasets. Our method aligns with this approach.

We first perform dense pixel tracking from the initial frame, producing dense 3D trajectories $\mathcal{P}^{(0,)} \in \mathbb{R}^{T \times H \times W \times 4}$. For any destination frame $\nu$, the 3D points $\mathcal{P}^{(0,\nu)} = \mathcal{P}^{(0,)}[\nu] \in \mathbb{R}^{H \times W \times 4}$ represent the 3D positions of all pixels from the initial frame in the camera coordinate system at frame $\nu$. This naturally combines the camera motion from frame 0 to $\nu$ with any non-rigid scene motion over the interval. This setup aligns with the methodology of MonST3R (Zhang et al., 2024). We then apply the global alignment approach from Wang et al. (2024b); Zhang et al. (2024) to estimate camera poses across dynamic video sequences.

In detail, we uniformly sample keyframes with a stride of 2 from the input video and densely track all pixels of these frames. For each keyframe $k$, tracking is performed both forward and backward within a window $[k - w, k + w]$, where $w = 8$. Following Zhang et al. (2024), we model the pose estimation as an optimization task, where the learnable parameters include *per-keyframe* depth maps, as well as *per-keyframe* intrinsic and extrinsic camera parameters, and optimize with gradient descent. The objective function combines alignment loss, temporal smoothness loss, and 2D flow loss. For the 2D flow loss, we calculate pseudo optical flow ground truth using our dense tracking approach rather than relying on an off-the-shelf model (Wang et al., 2025; Teed & Deng, 2020). For non-keyframe frames, camera poses are interpolated from the two nearest neighboring keyframes.

We evaluate the pose estimation performance on Sintel (Butler et al., 2012) and TUM-dynamics (Sturm et al., 2012) datasets. On Sintel, we follow the evaluation protocol in Zhang et al. (2024); Chen et al. (2024), which excludes static and easy scene, remaining 14 test sequences. For TUM-dynamics, we sample the first 90 frames with the temporal stride of 3. We report the Absolute Translation Error (ATE), Relative Translation Error (RPE trans), and Relative Rotation Error (RPE rot), after applying a Sim(3) Umeyama alignment on prediction to the ground truth. The results are reported in Table 10. Our method demonstrates comparable performance to state-of-the-art approaches specifically tailored for visual odometry and SLAM tasks. We visualize the camera pose and dynamic 3D scene reconstruction on casual videos in figure 9.

| Methods | Sintel | | | TUM-dynamics | | |
|---|---|---|---|---|---|---|
| | ATE↓ | RPE trans↓ | RPE rot↓ | ATE↓ | RPE trans↓ | RPE rot↓ |
| DROID-SLAM (Teed & Deng, 2021a) | 0.175 | 0.084 | 1.912 | - | - | - |
| DPVO (Teed et al., 2024) | 0.115 | 0.072 | 1.975 | - | - | - |
| ParticleSfM (Zhao et al., 2022) | 0.129 | 0.031 | 0.535 | - | - | - |
| LEAP-VO (Chen et al., 2024) | 0.089 | 0.066 | 1.250 | 0.068 | 0.008 | 1.686 |
| Robust-CVD (Kopf et al., 2021) | 0.360 | 0.154 | 3.443 | 0.153 | 0.026 | 3.528 |
| CasualSLAM (Zhang et al., 2022) | 0.141 | 0.035 | 0.615 | 0.071 | 0.010 | 1.712 |
| DUSt3R (Wang et al., 2024b) | 0.417 | 0.250 | 5.796 | 0.083 | 0.017 | 3.567 |
| MonST3R (Zhang et al., 2024) | 0.108 | 0.042 | 0.732 | 0.063 | 0.009 | 1.217 |
| Ours | 0.172 | 0.060 | 0.553 | 0.052 | 0.007 | 1.343 |

Table 10: **Pose estimation results** on Sintel and TUM datasets. The upper group (first four rows) includes methods that estimate camera poses only, without reconstructing scene geometry while the lower group and our approach provide both camera poses and per-frame depth maps. Our method achieves competitive results compared to other approaches specifically designed for visual odometry or SLAM tasks.

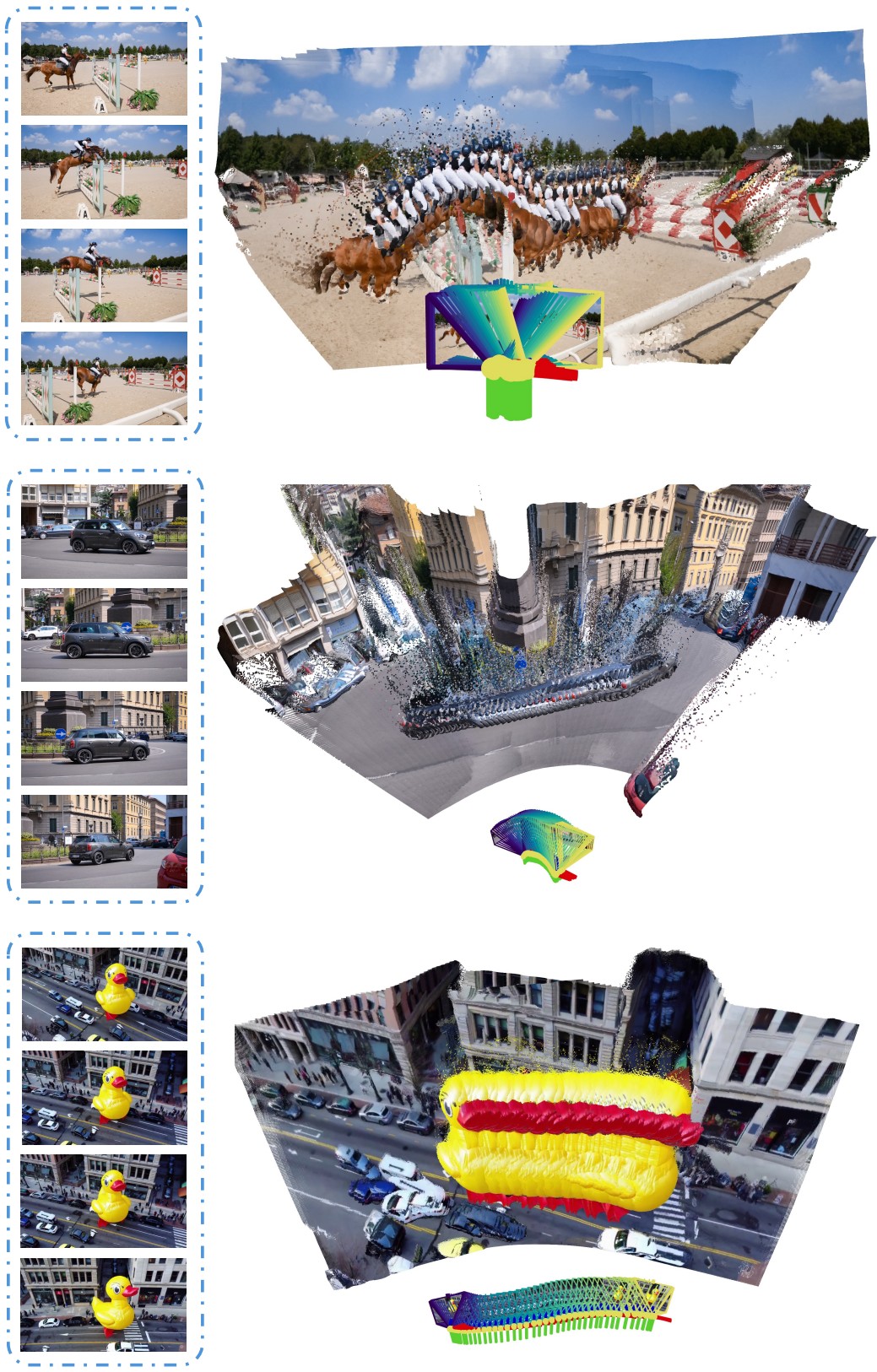

Figure 9: **Qualitative results of jointly depth and pose estimation** on in-the-wild videos (first two rows) and AI-generated video (last row) of our approach.

