# OpenReview forum: "DELTA: DENSE EFFICIENT LONG-RANGE 3D TRACKING FOR ANY VIDEO"
_ICLR.cc/2025/Conference — ICLR 2025 Poster_

### Official Review · Reviewer_oj5m · 2024-10-21

**Soundness:** 2
**Presentation:** 2
**Contribution:** 3
**Rating:** 6
**Confidence:** 5

**Summary:**

This paper proposes a method for dense 3D motion prediction, named TAPE3D. This paper finds that there is a redundant design in the CoTracker's global attention, and proposes their local-global attention, which reduces calculation costs to both accelerate the speed and reduce the resource requirement. After that, the paper proposes a transformer-based upsampler, which shows better performance than previous upsamplers, such as the ConvNet-based used in optical flow methods. This paper also finds that using log-depth as the depth representation is an optimal choice. Finally, TAPE3D achieves SoTA performance on both 2D/3D point tracking with the fastest speed.

Overall, TAPE3D is an interesting work that contributes some insights to the field, and this must be acknowledged. However, these contributions are a little lack depth, and feel somewhat scattered, like a collection of bundled findings. Considering that there are some parts not clearly written and are difficult to understand, I'm giving 'marginally below the acceptance threshold' for now.

**Strengths:**

1. This paper proposes a method, named TAPE3D, which not only obtains SoTA performance but also achieves the fastest speed.
2. This paper proposes the local-global cross-attenton, which reduces the calculation consumption while achieving better performance.
3. This paper proposes a transformer-based upsampler to upsample the sparse predictions to dense ones, achieving better performance than the ConvNet-based ones in previous optical flow methods.
4. This paper finds that using log-depth as the depth representation can obtain better performance.

**Weaknesses:**

1. Although many contributions are provided, they are a little lack depth, and feel somewhat scattered, like a collection of bundled findings. (but we have to acknowledge the contributions' value)
2. There are some parts that are not clearly written. In local-global attention, what N' (Fig 3) means, and why N' is different between training and evaluation? And it seems that there is no description of what M means (number of anchors if I understand correctly).
3. It seems that the differences between the proposed Local-Global attention and the global one in CoTracker are
    a) use anchor tracks to attend virtual tracks.
    b) add local attention.
Although a) can reduce the calculation consumption, b) needs additional calculation consumption. During evaluation, local-global even requires O(TKM + TLN^2) more calculation.
4. In L290-L295, I'm confused why the virtual tracks should "cross-attent to a local random patch of tracks". What is "local random patch of tracks"? In Fig.3, it seems that the virtual points only attend to anchor tracks in cross-attention and themselves in self-attention.
5. In Table.5, why BootsTAPIR+ZoeDepth is worse than CoTracker+ZoeDepth? As far as I know, BootsTAPTIR have much better performance than CoTracker in 2D point tracking, maybe more comparisons are required to show the reasonability (LocoTrack and TAPTRv2 for example).

**Questions:**

For the questions, please refer to the Weaknesses part. If the author can address my concerns, I will raise my score.

---

> ### Author Response · Authors · 2024-11-20
> **Response to Reviewer oj5m (1/2)**
>
> **Q. Although many contributions are provided, they are a little lack depth, and feel somewhat scattered, like a collection of bundled findings. (but we have to acknowledge the contributions' value)**
>
> Please see the answer 1 of our response to common issues.
>
> **Q. About the unclear writting**
>
> We re-wrote the Sec. 3.2 in our revised paper to improve clarity. Please see the answer 2 of our response to common issues.
>
> **Q. About the definition of M, N'? What is "local random patch of tracks"?**
>
> We have clarified the definition of M, N' in Sec 3.2 (L286-291) in our revised paper. In summary, during training, instead of using the entire frame as dense track, we randomly crop a small patch of size $N'=h' \times w'$ and supervise on this patch. Due to the fact that patchwise training limits spatial attention to the cropped region, neglecting the global context of the first frame, we introduce $M$ sparse anchor tracks, with starting positions uniformly sampled across the first frame. Thus, in the global attention, virtual tracks first cross-attend to these sparse anchors, followed by attention to the local patch. This strategy captures global motion while maintaining efficiency through reduced dense track size.
>
> **Q. It seems that the differences between the proposed Local-Global attention and the global one in CoTracker are a) use anchor tracks to attend virtual tracks. b) add local attention. Although a) can reduce the calculation consumption, b) needs additional calculation consumption. During evaluation, local-global even requires O(TKM + TLN^2) more calculation.**
>
> The computational cost for part (2) in Fig. 3 is actually $TK(N + N) + KT^2 + TK^2$, where $TK(N + N)$ represents the two cross-attention blocks. Initially, we used O notation, simplifying $O(TK(N + N))$ to $O(TKN)$. To avoid confusion, we’ve revised this notation to show the exact computational cost: $TK(N + N) + KT^2 + TK^2$. For our approach in part (3), the cost is $TK(N + 2M) + KT^2 + TK^2 + TLN$.
>
> When the number of anchor tracks $M$ is much smaller than the number of dense tracks $N$, we effectively reduce the global attention cost by $TKN$, while local attention adds only $TLN$. Since $L$ and $K$ are of a similar scale, our Local-Global attention has the same overall computational cost as CoTracker’s global-only attention but achieves significantly better performance in dense tracking.

---

> > ### Author Response · Authors · 2024-11-20
> > **Response to Reviewer oj5m (2/2)**
> >
> > **Q. In Table.5, why BootsTAPIR+ZoeDepth is worse than CoTracker+ZoeDepth? As far as I know, BootsTAPTIR has much better performance than CoTracker in 2D point tracking, maybe more comparisons are required to show the reasonability (LocoTrack and TAPTRv2 for example).**
> >
> > The performance of combined models like BootsTAPIR+ZoeDepth and CoTracker+ZoeDepth heavily depends on the **quality and scale-consistency of the input video depth**. However, frame-wise depth estimators such as ZoeDepth or Unidepth lack scale-consistency, as illustrated in the videos on our [anonymous website](https://tape3d.github.io/). Even small depth errors can result in significant 3D location discrepancy, meaning that better 2D tracking performance does not necessarily lead to better 3D tracking performance.
> >
> > We evaluated two additional 2D tracking models, LocoTrack and TAPTR, on the TAPVid-3D benchmark (noting TAPTRv2 is not yet open-source), as shown below. Similar behaviour is observed, where both models face limitations in 3D tracking.
> >
> > **Table R3**: **More quantitative results on the TAPVid3D benchmark.**
> > ||Aria | DriveTrack |Pstudio |Average |
> > |-|-|-|-|-|
> > || AJ$\uparrow$ / APD$\uparrow$ / OA$\uparrow$ |AJ$\uparrow$ / APD$\uparrow$ / $\uparrow$OA | AJ$\uparrow$ / APD$\uparrow$ / OA$\uparrow$ | AJ$\uparrow$ / APD$\uparrow$ / OA$\uparrow$ |
> > | LocoTrack+UniDepth | 15.1 / 24.0 / 83.5 | 13.0 /19.8 / 82.8 | 7.2 / 13.1 / 80.1 | 11.8 / 19.0 / 82.3
> > | TAPTR+UniDepth | 15.7 / 24.2/ 87.8 | 12.4 / 19.1 / 84.8 | 7.3 / 13.5 / 84.3 | 11.8 / 18.9 / 85.6
> > | **Ours**+UniDepth | 16.6 / 24.4 / 86.8 | 14.6 / 22.5 / 85.8 | 8.2 / 15.0 /76.4 | 13.1 / 20.6 / 83.0
> > |
> > | LocoTrack+ZoeDepth | 8.9 / 15.1 / 83.5 | 7.5 / 12.3 / 82.8 | 9.7 / 17.1 / 80.1 | 8.7 / 14.8 / 82.1 |
> > | TAPTR+ZoeDepth | 9.1 / 15.3 / 87.8 | 7.4 / 12.4 / 84.8 | 10.0 / 17.8 / 84.3 | 8.8 / 15.2 / 85.6
> > | **Ours**+ZoeDepth | 10.1 / 16.2 / 87.2 | 7.8 / 12.8 / 87.2 | 10.2 / 17.8 / 74.5 | 9.4 / 15.6 / 82.1
> >
> > Furthermore, 2D tracking + depth estimator can only predict motion for visible points within the viewpoint, restricting applications such as 3D/4D reconstruction, which require complete motion predictions, including occluded and out-of-frame regions (see the videos in our [anonymous website](https://tape3d.github.io/)).
> >
> > These challenges underline the need for **end-to-end 3D tracking models** to achieve reliable 3D performance. Our approach addresses these limitations and is also complementary to recent advances in 2D point tracking, enabling us to integrate their findings to further enhance both 2D and 3D tracking performance.
> >
> > **We would greatly appreciate it if you could let us know if the above answers can address your questions.**

---

> > > ### Comment · Reviewer_oj5m · 2024-11-25
> > >
> > > We appreciate the authors’ efforts in addressing my concerns, particularly the analyses provided in the second response. Will these analyses be included in the updated version? I think these analyses are interesting and the more comparisons (based on TAPTR & LocoTrack) can make Table 5 more solid.

---

> > > > ### Author Response · Authors · 2024-11-25
> > > > **Follow up with Reviewer oj5m**
> > > >
> > > > Thank you for your valuable feedback. In our latest revision, we have updated Table 5 to include results from two recent 2D tracking methods and added a detailed discussion  (see L898-912) in the Appendix. We greatly appreciate your time and effort in reviewing our work!

---

> > > > > ### Comment · Reviewer_oj5m · 2024-11-25
> > > > >
> > > > > Thank you again for your efforts during the rebuttal period! I think the authors have addressed all my concerns, and I'd like to raise my score to 6.

---

### Official Review · Reviewer_EecY · 2024-11-02

**Soundness:** 3
**Presentation:** 3
**Contribution:** 3
**Rating:** 6
**Confidence:** 3

**Summary:**

This paper presented a method, TAPE3D, that efficiently tracks every pixel of a frame throughout a video, demonstrating state-of-the-art accuracy in dense 2D/3D tracking while running significantly faster than existing 3D tracking methods.

**Strengths:**

1. TAPE3D introduces the spatial-temporal transformer to extract more visual features and uses the Upsampler module to obtain high-resolution results.
2. TAPE3D delivers dense 3D tracking at scale, running over 8x faster than previous methods while achieving state-of-the-art accuracy.

**Weaknesses:**

1. The idea of using the spatial-temporal transformer is not new, such as [1][2][3].
2. The authors are suggested to provide the compilation cost of each module to verify the efficiency of TAPE3D.
3. Are all inference experiments testing on a same machine? The information of the machine including GPU and CPU are suggested to provide.

[1] Hu M, Zhu X, Wang H, et al. Stdformer: Spatial-temporal motion transformer for multiple object tracking[J]. IEEE Transactions on Circuits and Systems for Video Technology, 2023, 33(11): 6571-6594.
[2] Zhang T, Jiao Q, Zhang Q, et al. Exploring Multi-modal Spatial-Temporal Contexts for High-performance RGB-T Tracking[J]. IEEE Transactions on Image Processing, 2024.
[3] Zhang T, Jin Z, Debattista K, et al. Enhancing visual tracking with a unified temporal Transformer framework[J]. IEEE Transactions on Intelligent Vehicles, 2024.

**Questions:**

See weakness. If questions addressd, I will increase the rating.

---

> ### Author Response · Authors · 2024-11-20
> **Response to Reviewer EecY**
>
> **Q. The idea of using the spatial-temporal transformer is not new.**
>
> We would like to argue that we did not claim the spatial-temporal transformer is our contribution. Our keys contributions are **the novel combination of global-local spatial attention** (Sec. 3.2), **the attention-based upsampler with Alibi** (Sec. 3.3), and **the log of depth change ratio representation** (Sec. 3.4).
>
> **Q. The authors are suggested to provide the compilation cost of each module to verify the efficiency of TAPE3D. Are all inference experiments testing on a same machine? The information of the machine including GPU and CPU are suggested to provide.**
>
> We thank the reviewer for this insightful suggestion. All computational cost experiments were conducted on a machine equipped with a single NVIDIA A100 80GB GPU and an Intel(R) Xeon® Platinum 8275CL CPU @ 3.00GHz. Below, we present a detailed profiling of each component in our approach as well as previous methods when densely tracking 384x512 pixels of a 100-frame video. We also include the performance (EPE) on long-range optical flow prediction on the CVO-extended set.
>
>
> **Table R2**: **Runtime analysis.**
> |**Method**|Per-pass Component Runtime$\downarrow$|Number of passes$\downarrow$|Total Runtime$\downarrow$|EPE$\downarrow$|
> |-|-|-|-|-|
> |CoTracker|Backbone: 0.155s; Transformer: 219.6s|4|880s|5.20|
> |SpaTracker|Backbone: 0.145s; Transformer: 62s|36|2270s|5.53|
> |SceneTracker|Backbone: 0.145s; Transformer: 30.6s|36|1100s|11.5|
> |DOT|Sparse Track: 30s; Optical flow: 5s|**1**|**35s**|4.98|
> |Ours| Backbone: 0.145s; Transformer: 114s; Upsampler: 0.6s | **1** | 115s |**3.67**|
>
>
> For the above profiling, we measure the runtime of the main components for each method. For CoTracker, SpaTracker, and SceneTracker, due to memory limitations, these methods cannot process the entire 384x512 tracks in a single forward pass. Instead, we set the number of tracks to the maximum that fits within the same amount of GPU memory and perform multiple passes until all pixels are tracked.
>
> DOT, being a two-stage approach, performs sparse tracking on approximately 8,000 points, followed by optical flow computation for each frame pair. We report the runtime of each stage accordingly. Although DOT demonstrates the lowest runtime, we observed a significant "flicker" effect in its optical flow predictions (see Fig. 6 and the videos in the [anonymous website](https://tape3d.github.io/)), caused by per-frame optical flow predictions.
>
> **We would greatly appreciate it if you could let us know if the above answers can address your questions.**

---

> > ### Author Response · Authors · 2024-11-25
> > **Follow up with Reviewer EecY**
> >
> > Dear Reviewer EecY,
> >
> > Thank you for your valuable comments to improve our work. As the rebuttal period is nearing its end, we would greatly appreciate your thoughts on our rebuttal and whether it addresses your concerns. Any feedback is welcome and much appreciated!
> >
> > Best,
> >
> > Authors

---

> > > ### Comment · Reviewer_EecY · 2024-11-26
> > >
> > > Thank you for your detailed response and for addressing the concerns raised in the review. I have increased my score to 6.

---

### Official Review · Reviewer_n65b · 2024-11-03

**Soundness:** 3
**Presentation:** 3
**Contribution:** 3
**Rating:** 6
**Confidence:** 5

**Summary:**

The paper proposes a model for tracking points in 3D densely, whose architecture is inspired by 2D sparse tracking CoTracker model, but with a number of modifications to 1) lift the tracks to 3D and 2) improve the model efficiency to allow for dense tracking with reasonable computational cost.

For lifting the tracks to 3D, the authors propose that the model takes RGB-D frames as input, and add depth tokens to the model's transformer framework (eq 1) to refine the depth predictions jointly with the 2D tracking predictions. This seems to be similar to the approach proposed in SceneTracker.

For tracking densely with reasonable computational cost, the authors adopt two different strategies. First, they perform most of the tracking prediction at a downsampled resolution (H/r, W/r) and propose a learnt cross-attention based upsampler to upsample the predictions to (H, W) resolution. Second, authors introduce some modifications to the spatial attention module proposed in CoTracker to increase efficiency. As in CoTracker, they perform cross-attention between virtual tracks and sparse tracks (here sampled on a uniform grid on the first frame and called "anchor tracks"), and then perform most of the computation by self-attention layers on the virtual tracks. However, they introduce a third tensor of "dense tracks", which they process by local attention and then fuse with the anchor tracks before decoding back the final refined dense tracks predictions by performing cross-attention with the virtual tracks.

The model is trained using a fully supervised setting by using synthetic videos generated with the Kubric simulator. The model is first pre-trained on the 2D tracking setting for 100k iterations, and then further trained on the 3D setting for an additional 100k iterations.

Their model is evaluated both for the 2D and 3D tracking tasks using standard benchmarks such as CVO, LSFOdyssey and TAPVid-3D. The results show the model obtains SOTA results on the CVO benchmark as well as in the TAPVid-3D benchmark. On the LSFOdyssey benchmark, the model trained on Kubric does not attain SOTA performance, but the one finetuned on LSFOdyseey does.

The authors conduct a series of ablation studies to show the impact of the design decisions regarding the depth respresentation, spatial attention module and upsampler.

**Strengths:**

This is one of the first papers addressing the TAP-3D problem with a feedforward model (not requiring test-time optimization), along with SceneTracker and SpatialTracker, which makes it a valuable contribution in this newly developing field.

In addition, the authors present quantitative results on both 2D tracking and 3D tracking which demonstrate the good performance of the model. The proposed model obtains SOTA results on the CVO benchmark as well as in the TAPVid-3D benchmark.

The visualizations included in the supplementary material help to further assess the quality of the 3D tracks produced by the model and compare it to the SceneTracker and SpatialTracker baselines. The results show a cleary superiority of TAPE3D over SceneTracker, and  slightly superior results compared to SpatialTracker. These observations are roughly in line to the reported quantitative results on TAPVid-3d.

The design choices proposed by the paper are thoroughly ablated in section 4.3 and Table 6, demonstrating the value of each of these design decisions.

Finally, Fig. 1 shows that in the dense-tracking scenario, TAPE3D is at least 8x faster than the baselines, while obtaining superior performance on TAPVid-3d.

**Weaknesses:**

While the paper is a valuable contribution with good results, the model design seems to be mainly a combination of ideas from SceneTracker and CoTracker. The paper does not clearly state which ideas are novel, and which are borrowed from these previous methods.

Furthermore, the experimental section only presents 2D tracking results on CVO, which is not the most widespread 2D tracking benchmark. Results on TAPVid-DAVIS would help in demonstrating the SOTA status of this model for 2D tracking.

However, the main weakness of the method lies in that some of technical contributions are not presented with sufficient clarity or mathematical rigor. First, it is not clear how the dense tracks from Fig. 3, (3), are computed and how they connect to the tokens G^i_t of eq (1). In addition, the eq. (3) seems mathematically incorrect or at least too vague in notation and lacking explanation. It is not explained whether the functions q and k are linear layers and on what they are applied. Also, it's not clear what the softmax function is applied over. Finally, this is referred to a as cross-attention, but the values seems to be missing from the equation. While Figure 4 helps in understanding the general idea of the proposed attention-based upsampling, it is not exactly clear how to connect it to eq. (3) and therefore to fully understand the proposed module.

Regarding the training procedure, it's not clear why the authors train on such small number of videos (5.6K) if these are synthetic videos that can be generated at a larger scale at ease. It's also not clear adding other data sources, such as PointOddysey would improve performance on the  TAPVid-3D benchmark.

**Questions:**

Could you please describe which elements in the model design are novel and which are borrowed from prior work?

Have you tried evaluating your method on TAPVid-DAVIS (first)? Does it obtain SOTA results on 2D tracking compared to CoTracker and BootsTAPIR?

Please clarify how the dense tracks in Fig. 3 (3) relate to the explanations in Sec. 3 and in particular to the tokens G^i_t of eq. (1).

Please clarify how the proposed attention-based upsampling works and provide a corrected eq. (3) if possible. Please explain how the \tau cross attention blocks from Fig. 4 operate. Are these multiple heads in parallel?

---

> ### Author Response · Authors · 2024-11-20
> **Response to Reviewer n65b (1/2)**
>
> **Q. While the paper is a valuable contribution with good results, the model design seems to be mainly a combination of ideas from SceneTracker and CoTracker. The paper does not clearly state which ideas are novel, and which are borrowed from these previous methods. Could you please describe which elements in the model design are novel and which are borrowed from prior work?**
>
> Please see the answer 1 of our response to common issues.
>
> **Q. Have you tried evaluating your method on TAPVid-DAVIS (first)? Does it obtain SOTA results on 2D tracking compared to CoTracker and BootsTAPIR?**
>
> We did include the 2D tracking performance on the TAPVid2D benchmark (covering DAVIS, Kinetics, and RGB-Stacking subsets) in Table 7 in the Appendix of the original version. We additionally include the performance of BootsTAPIR in our revised version. Our method achieves competitive results compared to concurrent methods specifically designed for 2D tracking. Notably, BootsTAPIR benefits from a large-scale dataset of 15M real-world videos to enhance sparse 2D tracking, whereas our approach focuses on the more challenging task of dense 3D tracking, trained only on synthetic data. We believe that fine-tuning our model on large-scale, real-world video data could further improve both its 2D and 3D tracking performance.
>
> **Q. It is not clear how the dense tracks from Fig. 3, (3), are computed and how they connect to the tokens $G^i_t$ of eq (1).**
>
> The dense tracks in Fig. 3 part (3) are initialized similarly to the sparse tracks in CoTracker, with the same initial position, depth value, and visibility. The only difference is the total number of tracks. In the sparse tracking setting, the number of tracks can be any arbitrary number but cannot encompass the entire frame's pixels, as this would significantly exceed GPU memory limits (80GB) for a single forward pass. In the dense tracking setting, however, we track every pixel in the reduced-resolution image, forming a dense grid of size $N=H/r \times W/r$ (See L.252-253 in our revised paper).
>
> Sparse tracks, dense tracks, and anchor tracks all are encoded using the eqn. (1) so that each track will be represented by a set of token $G$. The token $G^i_t$ represents a segment of the $i$-th track at time $t$ and serves as an abstract term that can refer to sparse tracks, dense tracks, or anchor tracks, depending on the context.
>
> **Q. The eq. (3) seems mathematically incorrect or at least too vague in notation and lacking explanation. Please clarify how the proposed attention-based upsampling works and provide a corrected eq. (3) if possible. Please explain how the $\tau$ cross attention blocks from Fig. 4 operate. Are these multiple heads in parallel?**
>
> Thank you for your constructive feedback. The equation 3 is not mathematically wrong, yet we have re-written it more clearly to avoid confusion.
>
> In the revised version, we have rewritten this section with more details (see Sec. 3.3 in the revised paper). Below, we briefly summary answer your concern:
> * **Feature map extraction**: We first extract a coarse resolution feature map $\mathcal{F}\_{coarse} \in \mathbb{R}^{ \frac{H}{r} \times \frac{W}{r} \times D}$  and a fine resolution feature map $\mathcal{F}\_{fine} \in \mathbb{R}^{H \times W \times D}$ with our convolution encoder and decoder.
> * **Input to Cross-Attention**: For each pixel $(u,v)$ in the fine-resolution map, its feature vector $\mathcal{F}\_{fine}^{(u,v)}$ serves as the *query*, while the corresponding $\kappa \times \kappa$ neighborhood in the coarse-resolution map $\{\mathcal{F}\_{\text{coarse}}^{(u'\_{j}, v'\_{j})}\}\_{j=1}^{\kappa \times \kappa} \in \mathbb{R}^{(\kappa \times \kappa) \times D}$ provides the *key* and *value*. Thus the fine-resolution feature map is refined by: $\mathcal{F}\_{fine}^{(u,v)} = CrossAttn\Big(\mathcal{F}\_{fine}^{(u,v)}, \{\mathcal{F}\_{coarse}^{(u'\_{j},v'\_{j})}\}\_{j=1}^{\kappa \times \kappa} \Big)$
>
> * **Operation of Cross-Attention**: The cross-attention operation is applied to every pixel in the fine-resolution map (see the revised text for the a more detailed form of the attention scores with Alibi). After this step, the refined fine-resolution feature map is passed through an MLP to predict the weight map $\mathcal{W} = MLP(\mathcal{F}_{fine})$, where $\mathcal{W} \in \mathbb{R}^{H \times W \times (\kappa \times \kappa)}$. This weight map enables us to compute high-resolution tracking by taking a weighted average of the coarse-resolution tracks.
> * **Multi-Head Cross-Attention Blocks**: In practice, we use a series of $\tau$ multi-head cross-attention blocks with Alibi bias [3].

---

> ### Author Response · Authors · 2024-11-20
> **Response to Reviewer n65b (2/2)**
>
> **Q. Regarding the training procedure, it's not clear why the authors train on such small number of videos (5.6K) if these are synthetic videos that can be generated at a larger scale at ease. It's also not clear adding other data sources, such as PointOddysey would improve performance on the TAPVid-3D benchmark.**
>
> While we could generate an unlimited number of synthetic videos, diversity is limited by the variety of scenes and objects in the Kubric simulation. Consistent with prior work [1, 2], we found that around 6K videos, enhanced with on-the-fly augmentations, provides adequate variations for training.
>
> We also experimented with combining Kubric and PointOdyssey synthetic data for training our 3D dense tracker. However, this reduced performance on the real-world test set (TAPVid3D) -- a similar behaviour is also observed in 2D tracking benchmarks (https://github.com/facebookresearch/co-tracker/issues/44#issuecomment-1938502621). Since both datasets are synthetic, combining them has limited impact on real-video performance. We believe that fine-tuning on appropriate real video datasets is essential to improve real-world 2D/3D tracking performance.
>
> **We would greatly appreciate it if you could let us know if the above answers can address your questions.**
>
> [1] Cotracker: It is better to track together.
>
> [2] Generative Camera Dolly: Extreme Monocular Dynamic Novel View Synthesis.
>
> [3] Train short, test long: Attention with linear biases enables input length extrapolation.

---

> > ### Author Response · Authors · 2024-11-25
> > **Follow up with Reviewer n65b**
> >
> > Dear Reviewer n65b,
> >
> > Thank you for your valuable comments to improve our work. As the rebuttal period is nearing its end, we would greatly appreciate your thoughts on our rebuttal and whether it addresses your concerns. Any feedback is welcome and much appreciated!
> >
> > Best,
> >
> > Authors

---

> > > ### Author Response · Authors · 2024-12-02
> > > **Follow up with Reviewer n65b**
> > >
> > > Dear Reviewer n65b:
> > >
> > > Thanks again for your constructive and insightful feedback to strengthen our work. As the rebuttal period is ending tomorrow, we wonder if our response answers your questions and addresses your concerns. Any feedback is welcome and much appreciated!
> > >
> > > Best,
> > >
> > > Authors

---

### Official Review · Reviewer_G77y · 2024-11-10

**Soundness:** 3
**Presentation:** 2
**Contribution:** 3
**Rating:** 6
**Confidence:** 3

**Summary:**

This paper presents a novel method for dense 3D tracking in monocular video. By introducing the anchor tracks and attention based upsampling methods, the proposed method significantly improves the inference speed by ~10x. By modifying the depth representation and supervision signal, the method significantluy improves the performance.

**Strengths:**

1. The proposed method indeed remedy the efficiency issue of previous SOTA methods significantly.
2. The experiments are comprehensive and convincing.

**Weaknesses:**

Some key concepts and design choices are not well explained.

1. Why do we need certain anchor tracks? What if we sample some certain points in the dense tracks?
2. The difference of sparse tracks and dense tracks are not well explained. For the comparisons in subfigure 2 and 3 in Fig.3, the dimension of sparse tracks and dense tracks are both T x N, then why it is called dense tracks? What is N' and L, the meaning of these two parameters are not well defined and explained.
3. If the depth is predicted from monocular video, the scale is not specified. How can we ensure their consistency across different frames?
4. The attention upsampling method is quite related to the well known guided filter work. It is better to discuss the relationship with it.

**Questions:**

1. Why the 2D version consistently better than 2D version in Table2? Is there any analysis on this unexpected results?
2. How the key parameters such as M, r, N' and L are chosen?

---

> ### Author Response · Authors · 2024-11-20
> **Response to Reviewer G77y (1/2)**
>
> **Q. Why do we need certain anchor tracks? What if we sample some certain points in the dense tracks?**
>
> Please see the answer 2 of our response to common issues.
>
> **Q. About the different between sparse and dense tracks**
>
> Please see the answer 2 of our response to common issues.
>
> **Q. What is N' and L, the meaning of these two parameters are not well defined and explained.**
>
> We have clarified the definition of N', L in Sec 3.2 in our revised paper. In brief,
> * During training, instead of using the entire frame as dense track, we randomly crop a small patch of size $N'=h' \times w'$ and supervise on this patch (L287).
> * To capture fine-grained representations of local relations among dense tracks, we apply self-attention within very small spatial patches containing L pixels (L300).
>
> **Q. If the depth is predicted from monocular video, the scale is not specified. How can we ensure their consistency across different frames?**
>
> Baseline methods that combine 2D track + depth estimators (e.g., CoTracker + ZoeDepth/UniDepth) heavily rely on the **quality and scale-consistency of the input video depth**. However, frame-wise depth estimators such as ZoeDepth or Unidepth lack scale-consistency, leading to inferior 3D tracking performance and the "jitter" effect (as illustrated in the videos on our [anonymous website](https://tape3d.github.io/)). In contrast, 3D tracking approaches like ours, SpaTracker, and SceneTracker are trained end-to-end for 3D tracking, enabling them to rectify depth across time even when the input depth video is inconsistent. As discussed in Sec. 3.4 of our paper, a key difference with prior work that further addresses this issue is to leverage a log-depth representation for depth, which is scale-invariant and reduces dependency on the depth map accuracy, thereby improving performance on real-world benchmarks where depth estimation is often inaccurate.
>
> Additionally, advancements in consistent video depth estimation methods, such as DepthAnything [1] or DepthCrafter [2], can further mitigate this challenge.
>
> **Q. The attention upsampling method is quite related to the well known guided filter work. It is better to discuss the relationship with it.**
>
> Our attention upsampler shares a conceptual similarity with guided filters in the use of a neighboring window in the low-resolution image as guidance. However, in our approach, we explicitly define each pixel’s output in the high-resolution map as a weighted combination of a neighboring region in the low-resolution map, with weights predicted directly through attention layers and an MLP. In this sense, the attention block acts as guidance in our case. The attention blocks are fully differentiable and optimized via gradient descent like guided filters.
>
> We also experimented with a baseline combining a convolution-based convex upsampler with a bilateral filter (a variant of the guided filter), but it did not yield significant improvements. Based on these observations, we argue that the attention-based upsampler is a more suitable and effective choice for this task, offering both flexibility and ease of learning.
>
> **Q. Why the 2D version is consistently better than 3D version in Table2? Is there any analysis on this unexpected results?**
>
> Our dense 2D tracking version performs slightly better than the 3D version in Table 2, though this difference is minor compared to the significant improvements that our method (both 2D and 3D) achieves over previous approaches. The slight drop in 2D performance for the 3D version can be attributed to the additional objectives involved in training, which include predicting 3D locations. To support this, the 3D model’s architecture incorporates depth correlation features and an additional head for predicting depth changes. These enhancements, while critical for 3D tracking, introduce slight trade-offs in optimizing pure 2D predictions.

---

> ### Author Response · Authors · 2024-11-20
> **Response to Reviewer G77y (2/2)**
>
> **Q. How the key parameters such as M, r, N' and L are chosen?**
>
> We empirically set these hyper-parameters to balance between the computational cost and model performance. More details regarding these parameters are included in the Appendix (L837-852).
>
> Below we add the ablation study on $L$ and $r$. These experiments are conducted with our 2D version.
>
> **Table R1.1**: **Ablation on the local patch size $L$.**
> ||L=0 (disable)|L=4*4|L=6*6|L=8*8|
> |-|-|-|-|-|
> |EPE $\downarrow$|3.75|3.69|**3.63**|3.70|
>
> Empirically we found that using patches of size $L=6\times 6$ in the local spatial attention yields the best performance.
>
> **Table R1.2**: **Ablation on the downsample factor $r$.**
> ||r=8|r=4|r=2|
> |-|-|-|-|
> |EPE $\downarrow$|3.98|3.63|**3.44**|
> |Runtime(s) $\downarrow$|**6.5**|10.4|67.3|
>
> A smaller upsample ratio ($r=2$) improves performance but increases runtime by about $6\times$, while a larger ratio ($r=8$) reduces performance. We select $r=4$ as a balance between performance and efficiency.
>
> We set $M=9 \times 12$ and $N'=30 \times 40$ during training to fit the model in the GPU memory.
>
> **We would greatly appreciate it if you could let us know if the above answers can address your questions.**
>
> [1] Depth anything: Unleashing the power of large-scale unlabeled data.
>
> [2] Depthcrafter: Generating consistent long depth sequences for open-world videos.
>
> [3] UniDepth: Universal Monocular Metric Depth Estimation.

---

> > ### Author Response · Authors · 2024-11-25
> > **Follow up with Reviewer G77y**
> >
> > Dear Reviewer G77y,
> >
> > Thank you for your valuable comments to improve our work. As the rebuttal period is nearing its end, we would greatly appreciate your thoughts on our rebuttal and whether it addresses your concerns. Any feedback is welcome and much appreciated!
> >
> > Best,
> >
> > Authors

---

> > > ### Author Response · Authors · 2024-12-02
> > > **Follow up with Reviewer G77y**
> > >
> > > Dear Reviewer G77y:
> > >
> > > Thanks again for your constructive and insightful feedback to strengthen our work. As the rebuttal period is ending tomorrow, we wonder if our response answers your questions and addresses your concerns. Any feedback is welcome and much appreciated!
> > >
> > > Best,
> > >
> > > Authors

---

### Author Response · Authors · 2024-11-19
**The General Response to Reviewers (1/2)**

We would like to thank all reviewers for their feedback. We are pleased that the reviewers acknowledged that: (i) our contributions to the 3D tracking are **"valuable and insightful for the field"** (Reviewer oj5m, n65b); (ii) our method achieves **significant improvements over recent approaches across multiple benchmarks while being 8x more efficient** (all reviewers); (iii) our paper includes **extensive experiments to justify our design choices** (Reviewer G77y).

**In the updated version of the paper, changes are marked in blue.** The changes to the paper are summarized as follows:
1. Sections 3.2 and 3.3 are modified to improve clarity.
2. Ablation on the anchor tracks (See Tab. 8 in the Appendix).
3. Qualitative comparisons of long-range optical flow prediction between our proposed attention-based upsampler and the CNN-based upsampler used in RAFT[4] are included (see Fig. 7 in the Appendix, and L908-912).
4. Qualitative comparisons of long-range optical flow prediction between our approach, DOT, and the SoTA optical flow method SEA-RAFT[3] are also included (See Fig. 8 in the Appendix, and L914-916).
5. A downstream application is added: non-rigid structure from motion from in-the-wild dynamic videos (See Fig. 9 in the Appendix, and L917).
6. Quantitative results of pose estimation on Sintel and TUM-dynamic dataset are added (see Tab. 9 in the Appendix).

**We also provide video demonstrations of our method on this [anonymous website](https://tape3d.github.io/)**, including:

1. More qualitative results on dense 3D tracking for in-the-wild videos.
2. More qualitative comparisons with 3D point tracking approaches (SpaTracker, SceneTracker, DOT3D) and lifted-3D versions of 2D point tracking methods (CoTracker and LocoTrack) on in-the-wild videos.
3. Demonstrations of the limitations of lifted-3D versions of 2D point tracking methods, particularly the "jitter" effect caused by inconsistent video depth and occlusion issues.
4. Qualitative results of non-rigid structure-from-motion.

---

> ### Author Response · Authors · 2024-11-19
> **The General Response to Reviewers (2/2)**
>
> Below, we address common questions raised by the reviewers.
>
> **Q1. Regarding the novelty of the method and each of the components.**
>
> Our paper tackles the critical yet underexplored task of dense, long-term, 3D tracking from in-the-wild videos—a capability vital for real-world applications such as robotics, augmented reality, and autonomous systems. Existing approaches struggle to simultaneously achieve **reliable, dense, long-range 3D** motion tracking **in a feed-forward manner**. As shown in Table 1, the closest comparable works, such as SceneTracker and SpatialTracker, are not designed for efficient dense tracking and are more than **eight times slower** than our approach. Our method is the first to overcome these limitations, achieving state-of-the-art performance across multiple tasks, including long-range 2D optical flow, dense 3D tracking, and 3D point tracking.
>
> We introduced three key technical designs that are essential for adapting recent advancements in 2D and 3D point tracking into an efficient, dense 3D tracker.
> 1. A combination of **global & local spatial attention** for tracking (Sec. 3.2) that captures both global and local spatial interaction for dense 3D tracking. This intuitive design choice has not been explored in motion tracking before.
> 2. An **attention-based upsampler enhanced with Alibi** (Sec. 3.3), to obtain high-resolution dense tracks with sharp motion boundaries. This new architecture demonstrates noticable improvements over commonly used convolution-based upsampler.
> 3. We found that using the **log of depth change ratio** representation (Sec. 3.4) achieves the best 3D tracking performance compared to other commonly used depth representations. While various depth representations, including log-of-depth, have been explored in the literature, there has been no comprehensive comparison to date. Our work is the first to provide a definitive conclusion in the field of long-term motion tracking.
>
> These contributions are carefully designed, well-motivated, intuitive, and essential for achieving our strong results, setting a new SoTA for dense, long-term 3D tracking.
>
> Beyond state-of-the-art tracking, we demonstrate broader applicability by extending our method to non-rigid structure-from-motion (see Appendix), enabling recovery of both the camera pose and dynamic 3D structure from monocular dynamic video — capabilities unattainable with existing tools like Colmap [1] or DUSt3R [2].
>
> We believe our contributions will inspire further research and applications in 3D/4D computer vision.
>
> **Q2. Regarding the definition of sparse, dense, and anchor tracks and the notation in Section 3.2: Joint Global-Local Attention**
>
> We thank the reviewers for highlighting the need for further clarification. In the revised version, we have improved these definitions for clarity. Please first refer to Sec. 3.2 for detailed explanation. Here we outline the key terminologies:
>
> * **Sparse track:** Used in methods like CoTracker and SpaTracker, sparse tracks refer to tracking a set of $N$ arbitrary points. Typically, $N$ ranges from 1 to 40K points, constrained by memory limitations.
>
> * **Dense track:** dense tracks refer to tracking every pixel from the first frame (or any other reference frame) across the entire video (see L149-L155: Problem Setup (Sec 3.)). In Fig. 3 (part 3), $N$ denotes the number of dense tracks, calculated as $N = H/r \times W/r$, as we track from a reduced-resolution image, then subsequently upsample to full resolution.
>
> * **Anchor track:** A set of $M$ tracks ($M \ll N$) uniformly sampled across the entire image. In our method, *anchor tracks* are auxiluary tracks which are introduced to efficiently compute *virtual tracks* through cross attention. This novel concept plays a crucial role in enabling efficient dense tracking and facilitating end-to-end training. (see L290-L299).
>
> We also clarify other important notations used in this section:
> * $N'$: during training, instead of using the entire reduced resolution ($H/r \times W/r$) as dense track, we randomly crop a small patch of size $N'=h' \times w'$ and supervise on this patch (see L286-288 in our revised paper).
> * $L$: refers to the patch size used in our local spatial attention mechanism (which is defined in the **Patch-wise dense local attention** in Sec 3.2), set to $L=6 \times 6$ in our experiments (see L300-305 in our revised paper).
>
>
> [1] COLMAP (https://colmap.github.io/)
>
> [2] Dust3r: Geometric 3d vision made easy.
>
> [3] Sea-raft: Simple, efficient, accurate raft for optical flow.
>
> [4] Raft: Recurrent all-pairs field transforms for optical flow.

---

### Meta-Review · Area_Chair_Ld5f · 2024-12-20

**Metareview:**

This paper receives comments from 4 reviewers which are all positive for the publication of this paper. The paper is well-written and contributes to the tracking community.

**Additional Comments On Reviewer Discussion:**

The authors have successfully addressed the issues raised by the reviewers, and all agree to the publication of the paper.

---

### Decision · Program_Chairs · 2025-01-22

Accept (Poster)